# Compute-Optimal Scaling for Value-Based Deep RL

**Preston Fu**[1,*]    **Oleh Rybkin**[1,*]    **Zhiyuan Zhou**[1]    **Michal Nauman**[1,2]
**Pieter Abbeel**[1]    **Sergey Levine**[1]    **Aviral Kumar**[3]

[1]UC Berkeley    [2]University of Warsaw    [3]Carnegie Mellon University

## Abstract

As models grow larger and training them becomes expensive, it becomes increasingly important to scale training recipes not just to larger models and more data, but to do so in a *compute-optimal* manner that extracts maximal performance per unit of compute. While such scaling has been well studied for language modeling, reinforcement learning (RL) has received less attention in this regard. In this paper, we investigate compute scaling for online, value-based deep RL. These methods present two primary axes for compute allocation: model capacity and the update-to-data (UTD) ratio. Given a fixed compute budget, we ask: how should resources be partitioned across these axes to maximize data efficiency? Our analysis reveals a nuanced interplay between model size, batch size, and UTD. In particular, we identify a phenomenon we call *TD-overfitting*: increasing the batch quickly harms Q-function accuracy for small models, but this effect is absent in large models, enabling effective use of large batch size at scale. We provide a mental model for understanding this phenomenon and build guidelines for choosing batch size and UTD to optimize compute usage. Our findings provide a grounded starting point for compute-optimal scaling in deep RL, mirroring studies in supervised learning but adapted to TD learning. Project page: `value-scaling.github.io`.

## 1 Introduction

Scaling compute plays a crucial role in the success of modern machine learning (ML). In natural language and computer vision, compute scaling takes a number of different forms: model size [19], the number of experts in a mixture-of-experts model [20], or test-time compute [46]. Since these approaches exhibit different opportunities and tradeoffs, a natural line of study has been to identify strategies for *"compute-optimal"* scaling [19], that prescribe how to allocate a given fixed amount of compute to attain the best downstream performance.

In this paper, we are interested in understanding tradeoffs between different ways to scale compute for value-based deep reinforcement learning (RL) methods based on temporal-difference (TD) learning to realize a similar promise of transforming more compute to better data efficiency. Value-based TD-learning methods typically provide two mechanisms to scale compute: first, increasing the capacity of the network representing the Q-function, and second, increasing the number of updates made per data point (i.e., the updates-to-data, UTD ratio) collected by acting in the environment. Scaling along these two sources present different benefits, challenges, and desiderata [34]. Therefore, in this paper, we ask: ***What is the best strategy to scale model size and UTD to translate a given fixed compute budget into maximal performance?***

Analogous to prior scaling studies in language models [19] and deep RL [39], addressing this question requires us to understand how scaling compute in different ways affects the behavior of the underlying TD-learning algorithm. Concretely, we will need a mental model of how scaling model size interacts with various other hyperparameters of the TD-learning algorithm, notably the UTD ratio. Most

---

*Equal contribution. Corresponding authors: `prestonfu@berkeley.edu`, `oleh.rybkin@gmail.com`, `aviralku@andrew.cmu.edu`. Code: `github.com/prestonfu/model_scaling`.

39th Conference on Neural Information Processing Systems (NeurIPS 2025).

prior work focuses on presenting a single performant set of hyperparameters, instead of providing an analysis to help obtain such a set [23, 34]; we start with a number of controlled analysis experiments.

Our analysis reveals several insights into the distinct, and perhaps even opposing, behavior of TD-learning when using small versus large model sizes. In contrast to supervised learning, where the largest useful batch size primarily depends on gradient noise and is otherwise independent of model size [30], we find that in TD-learning, smaller models perform best with small batch sizes, while larger models benefit from larger batch sizes. At the same time, corroborating prior work [39], we find that for any fixed model size, increasing the UTD ratio $\sigma$ reduces the maximally admissible batch size. To convert these observations into actionable guidance, we develop a mechanistic understanding of the interplay between batch size, model capacity, and UTD ratio, discussed in Section 5.

We observe that for any fixed UTD ratio, increasing batch size reduces training TD-error across all model sizes. However, generalization, as measured by the validation TD-error on a held-out set of transitions, is highly dependent on the model size. For small models, attempting to reduce the training TD-error with larger batch sizes leads to worse validation TD-error – a phenomenon we term *TD-overfitting*. In contrast, for large models, reducing training TD-error by increasing the batch size up to a threshold enables a lower validation TD-error. We trace the source of TD-overfitting to poor-quality TD-targets produced by smaller networks: updating to fit these targets can harm generalization on unseen state-action pairs. Empirically, we find that for each model size, there exists a maximal admissible batch size: further increasing the batch size to reduce the variance in the TD gradient amplifies overfitting. Equipped with this finding and the observation that high UTDs reduce the maximal admissible batch size, we prescribe a rule to identify optimal batch sizes for scaling up RL training under large compute budgets.

We then identify the best way to allocate compute between model size and the UTD ratio, given an upper bound on either compute or on a combination of compute and data budget. We obtain scaling rules that extrapolate to new budgets/compute for practitioners. Our contributions are:

- We analyze the behavior of TD-learning with larger models and observe that larger models mitigate a phenomenon we call *TD-overfitting*, where value generalization suffers due to poor TD-targets.

- Based on this analysis, we establish an empirical model of batch size given a UTD ratio and model size, and observe that larger models admit larger batch sizes.

- We provide an empirical model of jointly scaling UTD ratio and model size, and the laws for the optimal tradeoff between them.

## 2 Related Work

**Model scaling in deep RL.** While large models have been essential to many of the successes in ML [6, 7, 48, 4, 53], typical models used for standard state-based deep RL tasks remain small, usually limited to a few feedforward MLP layers [38, 17, 18]. This is partly because naïve model size scaling often causes divergence [2, 41, 34]. Previous works have shown that RL can be scaled to bigger models [52, 34, 25, 41, 24, 47] by using layer normalization [34], feature normalization [23, 26], or using classification losses [24, 8]. While these works focus on techniques that stabilize RL training, they do not investigate the relationship between model capacity and UTD. We leverage our proposed understanding of this relationship to achieve compute-optimal RL training. Furthermore, prior work considered the aspect of predictability in scaling model capacity in RL, but in the context of online policy gradients [15] or for RLHF reward model overparameterization [12]. In contrast, we study model scaling in value-based RL where gradients come from backpropagating the TD loss.

**Data and compute scaling in deep RL.** A considerable amount of research in RL focused on improving data efficiency through scaling the UTD ratio [3, 5] and find that one key challenge is overfitting [27, 3, 33]. Previous work reported mixed results with evaluating overfitting in online RL [22, 9], but we find validation TD-error to be predictive of TD-overfitting in our experiments, akin to Li et al. [27]. Our TD-overfitting analysis additionally contextualizes prior work showing that large batch sizes can degrade performance with small models [36] Prior works also considered scaling up data in parallelized simulations or world models for on-policy RL [31, 44, 40, 11, 45]. Instead, we focus on data-efficient off-policy learning algorithms and study resource allocation problems pertaining to compute allocations instead.

**Scaling laws in value-based RL.** Most extensions of scaling laws from supervised learning focus on language models and cross-entropy loss [19, 30, 32, 28], with few exceptions targeting downstream

metrics [10]. In contrast, off-policy RL involves distinct dynamics due to bootstrapping [9, 23, 29, 15], making direct transfer of supervised scaling laws unreliable. Prior work shows that scaling UTD in off-policy RL yields a peculiar law [39], but leaves model capacity unexplored. We extend this line of work by showing that off-policy RL scales predictably with both UTD and model size and in the process, uncover interesting insights about the interplay between batch sizes, overfitting, and UTD.

## 3   RL Preliminaries and Notation

In this paper, we study off-policy online RL, where the goal is to maximize an agent's return by training on a replay buffer and periodically collecting new data [49]. Value-based deep RL methods train a Q-network, $Q_\theta$ by minimizing the temporal difference (TD) error:

$$L(\theta) = \mathbb{E}_{(s,a,s')\sim\mathcal{P},a'\sim\pi(\cdot|s')} \left[ \left( r(s,a) + \gamma\bar{Q}(s',a') - Q_\theta(s,a) \right)^2 \right], \tag{3.1}$$

where $\mathcal{P}$ is the replay buffer, $\bar{Q}$ is the target Q-network, $s$ denotes a state, and $a'$ is an action drawn from a policy $\pi(\cdot|s)$ that aims to maximize $Q_\theta(s,a)$. The ratio of the number of gradient steps per unit amount of data is called the *UTD ratio* (i.e., the updates-to-data ratio) and we will denote it as $\sigma$.

## 4   A Formal Definition of Compute-Optimal Scaling

Our goal in this paper is to develop a prescription for allocating a fixed compute budget or a fixed compute and data budget, between scaling the model size and the update-to-data (UTD) ratio for a value-based RL algorithm. As mentioned earlier, scaling model size and increasing the UTD ratio involve different trade-offs in terms of computational cost and practical feasibility. For example, scaling the UTD ratio results in more "sequential" computation for training the value function, which in turn implies a higher wall-clock time but does not substantially increase GPU memory needed. On the other hand, increasing model size largely results in more parallel computation (unless the model architecture itself requires sequential computation, a case that we do not study in this work). Answering how to best partition compute between the UTD ratio and model size enables us to also build an understanding of sequential vs parallel compute for training value functions. In this section, we formalize this resource allocation problem, building upon the framework of Rybkin et al. [39].

To introduce this resource allocation problem, we need a relationship between the compute $\mathcal{C}_J$ and the total data $\mathcal{D}_J$ needed to attain a given target return value $J$, the model size $N$, and the UTD ratio $\sigma$. Formally, we can represent the total compute in FLOPs as follows [39]:

$$\mathcal{C}_J(\sigma, N) \propto \sigma \cdot N \cdot \mathcal{D}_J(\sigma, N), \tag{4.1}$$

where $\mathcal{D}_J(\sigma, N)$ denotes the total amount of samples needed to attain performance $J$, and $\mathcal{C}_J(\sigma, N)$ denotes the corresponding compute. Since batch size is typically parallelizable, does not significantly affect wall-clock time, and is typically much smaller than the replay buffer, we drop the dependency of compute on the batch size and aim to optimize compute per unit datapoint. Finally, we denote the performance of a value-based RL algorithm Alg as $J(\pi_{\text{Alg}})$. With these definitions, we formalize:

---

**Problem 4.1** (Compute allocation problem). Find the best configuration for the UTD ratio $\sigma$ and the model size $N$, such that algorithm Alg attains:

1. *Maximal compute efficiency in attaining performance $J_0$ given data budget $\mathcal{D}_0$:*
$$(\sigma^*, N^*) := \arg\min_{(\sigma,N)} \quad \mathcal{C} \ \text{ s.t. } \ J\left(\pi_{\text{Alg}}(\sigma, N)\right) \geq J_0, \ \mathcal{D} \leq \mathcal{D}_0$$

2. *Maximal performance given budget $\mathcal{F}_0$ and coefficient $\delta$ for trading off compute/data:*
$$(\sigma^*, N^*) := \arg\max_{(\sigma,N)} \quad J\left(\pi_{\text{Alg}}(\sigma, N)\right) \ \text{ s.t. } \ \mathcal{C} + \delta\mathcal{D} \leq \mathcal{F}_0.$$

---

**The first part** of Problem 4.1 seeks to allocate a compute budget $\mathcal{C}_0$ between $N$ and $\sigma$ to minimize compute required to reach return $J_0$. From a practitioner's perspective, the solution to this part should prescribe how to attain a target level of performance as efficiently as possible given a certain amount of GPU resources available for training. **The second part** aims to construct a law that extrapolates to higher compute budgets and higher return. Instead of extrapolating as a function of return, which can be arbitrary and not predictable, we follow Rybkin et al. [39] and extrapolate as a function of a *budget* $\mathcal{F} = \mathcal{C} + \delta \cdot \mathcal{D}$. This allows the practitioners to achieve optimal return given the budget of resources available, where $\delta$ denotes the cost of data relative to compute, expressed e.g. in wall-clock time.

**Experimental setup.** We use BRO [34] and SimbaV2 [26], approaches based on SAC [13] that use a regularized residual network to represent the Q-values and have been shown to scale well to high capacities. These prior works showed that scaling width gives better performance that scaling depth in TD-learning. Thus, to study the impact of model size, we vary only the network width in $\{256, 512, 1024, 2048, 4096\}$. We consider batch sizes from 4 to 4096 (varied in powers of 2 and UTD ratios of 1, 2, 4, 8. We keep other hyperparameters fixed across all tasks at values suggested by Nauman et al. [34]. For our initial study, we leverage the results from [34] on Deepmind Control suite [50]. Following prior work [14, 34], we separate these into 7 medium difficulty tasks (referred to as DMC-medium) and 6 hard difficulty tasks (DMC-hard). For these tasks, we fit averages of the tasks for the two suites respectively, building upon the protocol prescribed in Rybkin et al. [39], to show generalization of our fits across tasks. We evaluate scaling on 4 more difficult tasks from DMC and HumanoidBench [42], where we make fits for each task individually to show applicability to single tasks. Further details are in Appendix B.

## 5 Analyzing the Interplay Between Model Size and Batch Size

Rybkin et al. [39] argues that the best batch size decreases as a power law with respect to the UTD ratio $\sigma$. However, this prior analysis holds model size $N$ constant and does not consider its influence on batch size. We extend prior analysis [39] by considering how model size modulates the effective batch size under fixed UTD ratio, revealing a distinct form of overfitting unique to TD-learning.

### 5.1 Measuring Overfitting in TD-Learning

Following Rybkin et al. [39], which identifies overfitting as a key factor in selecting effective batch sizes for a fixed model size, we begin our analysis by understanding how overfitting depends on model size. Unlike supervised learning, where the target is fixed, TD-learning involves fitting to targets that evolve over time and depend on the network being trained. This makes overfitting in TD-learning fundamentally different. As a measure of generalization, we measure the TD-error on both the training data (i.e., transitions sampled from the replay buffer) and a held-out validation set of transitions drawn i.i.d. from the same distribution. Further details are provided in Appendix B.

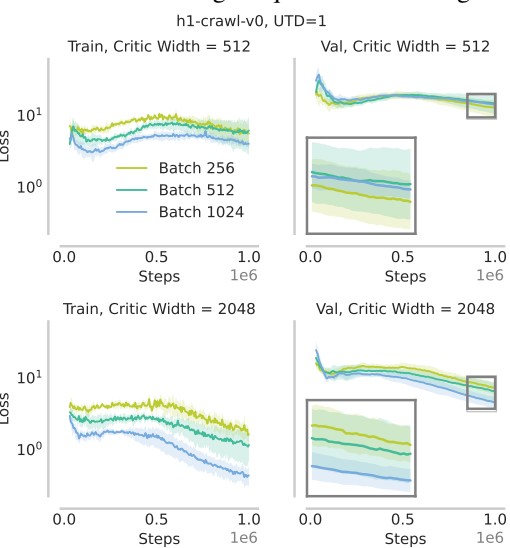

Figure 1: **Measuring train and validation TD-errors for different batch sizes** on `h1-crawl`. While the training and validation TD-errors reduce as model size increases, for smaller models a larger batch size results in a higher final TD-error. This illustrates the role of batch size in modulating overfitting with TD-learning.

***Observations on model size.*** We report training and validation TD-errors on `h1-crawl` at the end of training in Figure 2(a) (see Appendix D.5 for complete loss curves). As model size increases, the final training TD-error decreases, consistent with increased model capacity. Interestingly, we find that increasing model capacity consistently leads to a lower validation TD-error. Moreover, there is no clear sign of classical overfitting (i.e., low training error but high validation error), perhaps because TD-learning rarely "fully" fits target values regardless of model size.

***Observations on batch size.*** We next study the role of batch size in Figure 1 (when varying batch sizes for a fixed model size) and Figure 2(b, c). Perhaps as expected, larger batch sizes generally reduce training TD-error, likely because they provide a better low-variance estimate of the gradient. However, their impact on validation TD-error is more nuanced and depends on the model size $N$. For smaller networks (widths $\{256, 512\}$), increasing the batch size often plateaus or increases the validation TD-error. This corroborates prior work [39], which identified larger batch sizes as a source of overfitting when operating at networks with width 512. However, larger models allow us to use larger batch sizes without overfitting (Figure 2(d)). Why does this occur?

### 5.2 A Mental Model for TD-Overfitting

In supervised learning, overfitting occurs when reducing training loss further would primarily fit to noise or spurious correlations on the training dataset, in a way that results in a higher loss on a

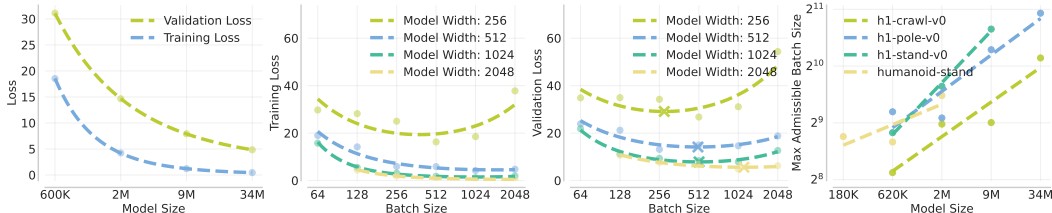

Figure 2: **Effect of batch size on TD-error for** `h1-crawl` **with** $\sigma = 1$. Left to right: **(a)** increasing model size consistently lowers the best achieved validation TD-error for a fixed batch size; **(b)** Larger batch sizes reduce training TD-error. **(c)** However, beyond a certain threshold, larger batch sizes lead to increased validation TD-error, particularly for smaller models, indicating TD-overfitting. **(d)** This overfitting threshold increases with model size: larger models can enable higher batch sizes, suggesting increased robustness to overfitting.

validation dataset distributed identically as the training data. Even though smaller networks overfit (Figure 2(c)), our experiments are not in this regime since larger networks are able to attain *both* lower training TD-error and lower validation error (Figures 2(b, c)).

We argue that this apparent deviation from classical overfitting is explained by the use of target networks. Regardless of whether a given network has sufficient capacity to reduce TD error on the current batch, TD methods would subsequently update the target network. This can lead to an increase in TD-error on validation data at the next step. That is, TD-error may not reduce: **(i)** on validation state-action pairs or **(ii)** with respect to updated target values.

For conceptual understanding: low-capacity Q-networks entangle features used to predict Q-values across state-action pairs [23, 21]. Target network updates inevitably change target values on unseen transitions, potentially increasing the validation TD-error, as we observe empirically in Figure 2(b, c) (full curves in Figure 13). Larger batch sizes produce lower-variance gradients that exacerbate this problem, as fitting the targets on some transitions comes at the expense of others with limited representational capacity.

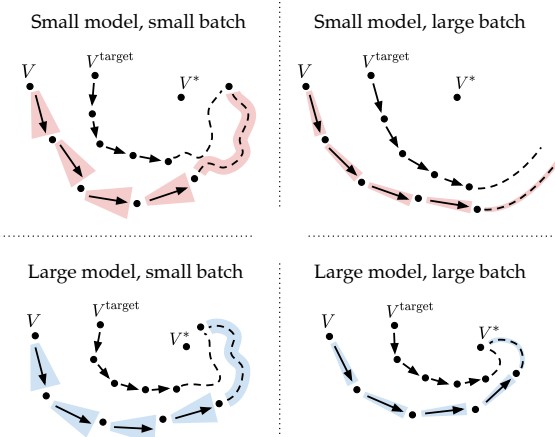

Figure 3: **A conceptual view of TD-overfitting.** Small models cannot cope with large batch sizes due to more directed gradient updates onto low-quality TD-targets, and might diverge from the target optimal value function $V^*$. Instead, they might perform better with smaller batch sizes, which result in noisy updates. Large models produce TD targets that are high-quality and benefit from regressing to these targets better via larger batch sizes.

In contrast, larger-capacity models can more effectively decouple predictions across transitions, mitigating this issue and leading to improved generalization even at high batch sizes. This suggests a key observation: *avoiding overfitting in TD-learning requires either smaller batch sizes or higher model capacity.* We present this insight as an illustration in Figure 3. We note that high capacity model generally leads to lower training and validation TD-errors (Figure 2(b, c)). We term this phenomenon **TD-overfitting**, to emphasize that it is driven not by memorization of values on the training set but by the interplay between limited capacity and non-stationary targets, that exist uniquely in TD-learning.

> **Takeaway 1: Smaller models cannot utilize large batch sizes**
>
> The training TD-error decreases with higher batch size. With low model capacity, increasing batch size results in a higher validation TD-error, i.e. the maximum admissible batch size is small. Larger models enable the use of a larger admissible batch size. This can be attributed to poor generalization of the TD-targets from smaller models.

## 5.3   The Role of Dynamic Programming in TD-Overfitting

We now conduct experiments to verify that indeed TD-targets are the root cause of the TD-overfitting phenomenon discussed above. To this end, we conduct a diagnostic experiment inspired

by the setup in Ostrovski et al. [37]. We train a *passive* critic alongside the main Q-network. This passive critic is not involved in TD updates but is trained solely to fit the TD-targets generated by the main Q-network (experimental details in Appendix B). By ablating the network width used to parameterize the passive critic, we can decouple the contributions of **(i)** the main Q-network's capacity and **(ii)** the quality of TD-targets, which we hypothesize drives TD-overfitting.

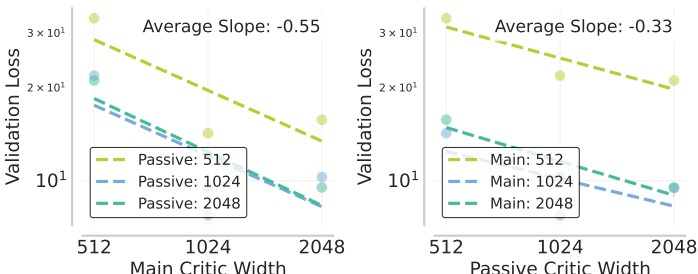

Figure 4: **Validation TD-error w/ the passive critic**. Increasing the model size for both the main and passive critic can reduce the validation TD-error, increasing the main critic size is much more effective, showing that target quality is crucial for effective learning.

In Figure 4, we observe that when the TD-targets are generated by a low-capacity Q-network, the passive critic — regardless of its own capacity — exhibits higher validation TD-error. *This supports the main insight from our mental model:* the TD overfitting phenomenon is driven primarily by the poor generalization of TD-targets produced by low-capacity networks. While increasing the passive critic's width improves its ability to fit lower quality targets (e.g., a width = 2048 passive critic fits width = 512 targets slightly better than a smaller passive critic), the validation TD-error can increase or slowly decrease over the course of training (see `h1-crawl` in Figure 14). Conversely, when the main Q-network generating the targets is larger (e.g., width = 2048), even smaller passive critics (e.g., width = 512 or 1024) can match resulting TD-targets quite well, and validation error decreases over the course of training (see `h1-crawl` in Figure 14). This indicates that a large portion of the overfitting dynamics of TD-learning is governed by the target values, and how they evolve during training, irrespective of the main critic (though the targets themselves depend on the main critic).

We also observe that if the passive critic is smaller than the target-network, it may underfit in its ability to fit the TD-targets on the training data, leading to elevated training and validation TD-errors. However, this underfitting effect is much less severe than the overfitting observed when the TD-targets themselves are poor due to the limited capacity of the Q-network, as seen from the slope in Figure 4.

---

**Takeaway 2: Overfitting in TD-learning is governed by TD-targets**

Overfitting in TD-learning is less about fitting the TD-targets on limited data, but more about the quality of the TD-targets themselves — a direct consequence of model capacity and the fundamental nature of dynamic programming with deep neural networks.

---

## 6 Prescribing Batch Sizes Using Model Size and the UTD ratio

Using the insights developed above, we now construct a prescriptive rule to select effective batch sizes, which in turn allows us to estimate the tradeoffs between UTD ratio and compute in the next section for addressing Problem 4.1. Specifically, we aim to identify the largest batch size, denoted $\tilde{B}$, that can be used before the onset of TD overfitting, i.e., before validation TD-error begins to increase for a given model size. From our insights in the previous section, we see that the largest such value of $\tilde{B}$ increases as model size $N$ increases. We also note in Figure 5, that $\tilde{B}$ decreases with increasing UTD ratio $\sigma$, aligned with the findings of Rybkin et al. [39]: for a fixed model size, larger $\sigma$ values lead to TD-overfitting when batch size increases. Motivated by these empirical findings, we propose a rule that models $\tilde{B}$ asymmetrically between model size $N$, and UTD ratio $\sigma$ as follows:

$$\tilde{B}(\sigma, N) \approx \frac{a_B}{\sigma^{\alpha_B} + b_B \cdot \sigma^{\alpha_B} \cdot N^{-\beta_B}} \tag{6.1}$$

where $a_B, b_B, \alpha_B, \beta_B > 0$ take on values listed in Appendix D.1.

---

**Scaling Observation 1: Batch size selection**

For best performance, batch size should increase with model size and decrease with the UTD ratio. This dependency can be modeled by a predictable function in Eq. (6.1).

---

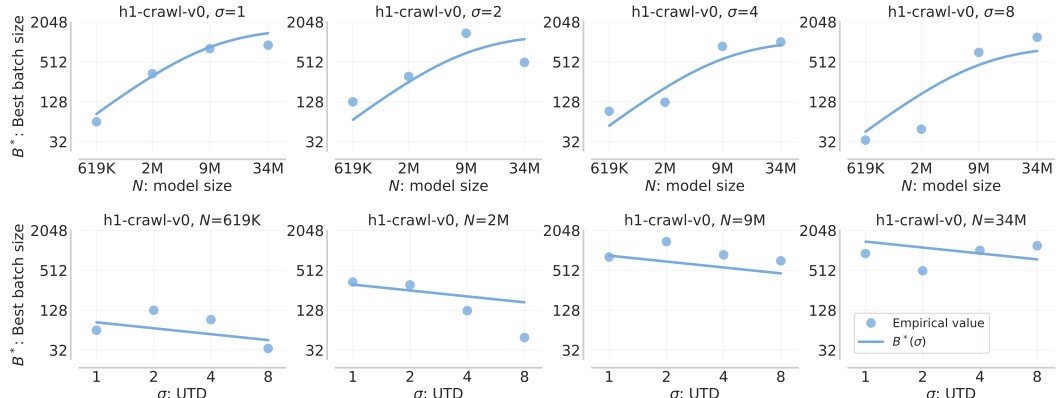

Figure 5: **A two-dimensional batch size fit** $\tilde{B}(\sigma, N)$. Slices of the fit are shown at particular values of $\sigma$ and $N$. In line with our analysis of TD-overfitting, we observe that larger models allow larger batch sizes. We build a fit that captures the intuition, but this effect does not continue indefinitely but instead asymptotes. We further extend prior work that observed that batch size needs to be decreased with UTD [39] and incorporate that in our 2-dimensional fit. Leveraging batch sizes from this fit allows us to better answer compute allocation questions.

**Implications.** As $\sigma$ increases, the $\sigma^{\alpha_B}$ term in the denominator dominates, yielding the approximation $\tilde{B}(\sigma, N) \approx a_B/\sigma^{\alpha_B}$, consistent with a power law relation from prior work [39]. Conversely, as $\sigma \to 0^+$, i.e., when targets are nearly static, $\tilde{B} \to \infty$ since updates are infrequent and TD-overfitting does not occur. Finally, our functional form increases with $N$, with an upward asymptote at $B^* \to a_B/\sigma^{\alpha_B}$ when $N \to \infty$. This reflects the intuition that low-capacity networks require smaller batch sizes to avoid TD-overfitting, whereas for sufficiently large models, the maximum admissible batch size is primarily constrained by the UTD ratio $\sigma$.

Crucially note that our proposed functional form factorizes into a function of $\sigma$ and a function of $N$,

$$\frac{a_B}{\sigma^{\alpha_B}} \cdot \frac{1}{1 + b_B \cdot N^{-\beta_B}}. \tag{6.2}$$

To evaluate the feasibility of such a relationship in our analysis, we had to run a 3-dimensional grid search $B \times N \times \sigma$. However, the fact that this fit is effective subsequently allows practitioners to instead run two 2-dimensional grid searches on $B \times \sigma$ and $B \times N$, significantly reducing the amount of compute needed to estimate this fit.

**Evaluation.** In order to evaluate this fit, we compare to a simple log-linear fit $\log \tilde{B} \sim (1, \log \sigma, \log N)$, which would increase indefinitely in $\sigma$ and $N$. Averaged over 4 tasks, our rule achieves a relative error of 48.9% compared to the log-linear fit's 55.1% (see also Appendix D). One might wonder if a relative error of 48.9% is actually large for a batch size fit. We empirically observe that the error is large because, in many cases, there is a wide range of batch sizes that all attain good performance (see Appendix D.2).

## 7 Partitioning Compute Optimally Between Model Size and UTD

Equipped with the relationship between batch size, model size, the UTD ratio from Eq. (6.1), and the definition of compute in Eq. (4.1), we now answer the questions from Problem 4.1.[1]

### 7.1 Solving Problem 4.1, Part 1: Maximal Data Efficiency under Compute $\mathcal{C}_0$

To solve this problem, we note that to maximize the data-efficiency we should operate in a regime where the total compute, $\mathcal{C} := k \cdot \sigma \cdot N \cdot \mathcal{D}_J(\sigma, N) \leq \mathcal{C}_0$, where $\mathcal{C}_0$ and $k$ are constants not dependent on $\sigma$ and $N$. We then require a functional form for $\mathcal{D}_J$. We observe that extending the relationship from Rybkin et al. [39], which modeled data efficiency as an inverse power law of the UTD ratio for a fixed, given model size, can also work well in our scenario when the model size is variable. Inspired by prior work on language models [19, 43], we augment the fit from Rybkin et al. [39] with an additive power law term dependent on model size. Intuitively, this is sensible since the total amount of data needed to attain a given performance should depend *inversely* on the model size since

---

[1]We also attempted to build a fit for learning rate in our experiments, but found learning rate to generally not be as critical as the batch size for compute-optimal scaling. Please see Appendix D.3 for more details.

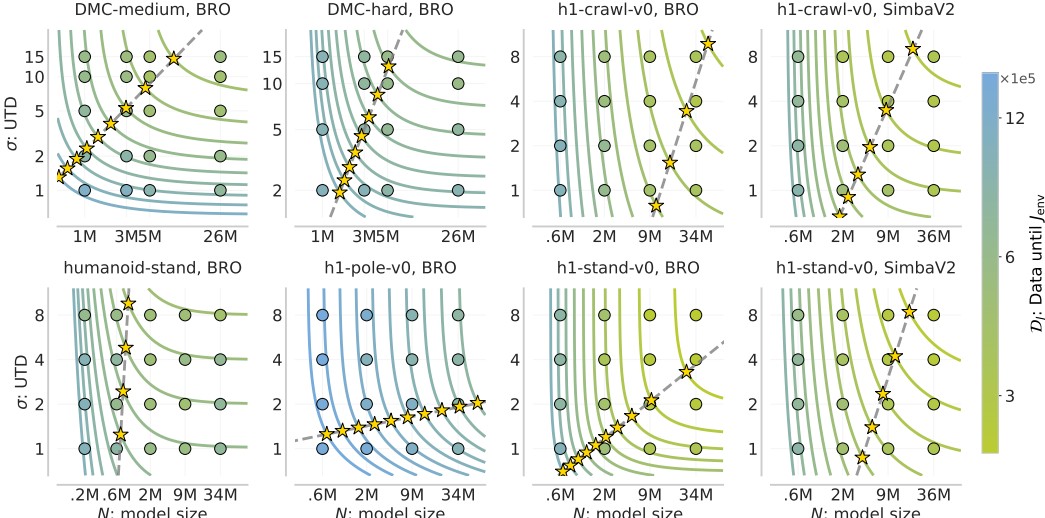

Figure 6: **Data efficiency fit $\mathcal{D}_J(\sigma, N)$ on all domains, shown as iso-data contours.** Each contour denotes the curve which attains the same data efficiency to attain a given target performance $J$, with data efficiency denoted by color. The form of the fit allows a closed-form solution for optimal configurations, and we show these as stars. These points lie on a power law. This law enables us to scale compute while allocating it to UTD or model size as we will discuss in subsequent results in this paper.

bigger models attain better sample-efficiency in TD-learning [34, 25]. Moreover, model size and the UTD ratio present two avenues for improving data efficiency, hence the additive nature of the proposed relationship. We find that the relationship can be captured as:

$$\mathcal{D}_J(\sigma, N) \approx \mathcal{D}_J^{\min} + \left(\frac{a_J}{\sigma}\right)^{\alpha_J} + \left(\frac{b_J}{N}\right)^{\beta_J}, \tag{7.1}$$

where $\mathcal{D}_J^{\min}$ is a constant not dependent on $\sigma$ and $N$, and $a_J, \alpha_J, b_J, \beta_J$ are constants that depend on the return target $J$. With this relationship in place, we are now able to answer Part 1:

$$\sigma^*(\mathcal{D}_0) = \left(\frac{a_\sigma}{\mathcal{D}_0 - \mathcal{D}^{\min}}\right)^{\alpha_\sigma}, \quad N^*(\mathcal{D}_0) = \left(\frac{b_N}{\mathcal{D}_0 - \mathcal{D}^{\min}}\right)^{\beta_N}, \tag{7.2}$$

where the coefficients can be computed from $a_J, \alpha_J, b_J, \beta_J$ (see details in Appendix A).

> **Scaling Observation 2: Partitioning compute optimally between model size and UTD**
>
> Optimal UTD and model size is a predictable function of data budget $\mathcal{D}$ (alternatively, compute budget $\mathcal{C}$), as a power law in Eq. (7.2).

We visualize this solution in Figure 6. We plot iso-$\mathcal{D}$ contours, i.e. curves in $(\sigma, N)$ space that attain identical data efficiency, and find that these curves move diagonally to the top-right for smaller $\mathcal{D}$ values, in a way where both increasing the model size and the UTD ratio improves data efficiency. These contours are curved such that there is a single point on each frontier that attains optimal compute efficiency $\mathcal{C}$. We plot these points, which follow the solution in Eq. (7.2). This allows us to predict data efficiency for novel combinations of UTD and model size, which is crucial.

**Evaluation.** Our data efficiency coefficients are fitted against a grid of UTD ratios and model sizes. We evaluate our proposed data efficiency fits on a grid of interpolated and extrapolated UTD ratios and model sizes using the fitted batch size. Averaged over 4 tasks, our fit achieves a relative error of 10.0% against the ground truth data efficiency on fitted UTD ratios and model sizes, 14.9% on interpolation, and 18.0% on extrapolation. Experimental details are described in Appendix B.1.

We also compare our estimated UTD ratios and model size with other approaches for allocating unseen compute budgets in Table 1. We compare to the following alternate approaches: **(i)** *UTD-only scaling* at compute budget $\mathcal{C}$ for a given model size, **(ii)** *model-only scaling* at compute budget $\mathcal{C}$ for a given UTD, and **(iii)** our proposed compute-optimal UTD and model size, run with a constant, fixed batch size not specifically designed for our compute budget $\mathcal{C}$. This constant fixed batch size corresponds to

the batch size prescribed by our fit for the first compute budget $\tilde{B}(\sigma^*(\mathcal{C}_{\min}), N^*(\mathcal{C}_{\min})))$. In Table 1, we observe that our compute-optimal scaling achieves the target performance using the least amount of data, whereas both $\sigma$-only scaling and $N$-only scaling require substantially more data, as evaluated using the ratio of the total amount of data needed for the approaches and the total amount of data needed for our compute-optimal approach. The strategy of using a constant batch size performs only marginally worse than our approach. However, as this comparison still relies on our proposed UTD ratio and model-size prescriptions, it primarily shows that these prescriptions are relatively robust to variations in batch size.

**Implications.** Our results show that appropriate choices of UTD and model size improve both data efficiency and compute utilization. At the same time, we find broad regions of near-equivalent performance: multiple (UTD, model-size) settings perform similarly well, so fully optimizing these hyperparameters is often unnecessary to capture most of the gains (Figure 20 and Figure 21). Similarly, while the best configuration is environment-dependent, with some tasks benefiting from larger models to begin learning and others from a higher UTD, **scaling the model size paired with a mild increase in UTD is often a good starting point.** Our framework makes these trade-offs explicit and provides a principled approach to selecting good values for these hyperparameters.

Table 1: **Data efficiency ratios** of various approaches to allocate compute to our approach of compute-optimal $(\sigma, N)$ scaling. All perform subpar to our compute-optimal UTD, model size prescriptions in terms of data efficiency.

| Approach | Mean | Median |
|---|---|---|
| Compute-optimal (ours) | 1.00 | 1.00 |
| Compute-optimal (ours) + fixed batch size | 1.03 | 1.05 |
| $\sigma$-only scaling | 1.26 | 1.18 |
| $N$-only scaling | 1.11 | 1.11 |

## 7.2  Solving Problem 4.1, Part 2: Resource Partitioning for Different Returns $J$

For the solution to the problem to be practical, we need to prescribe a solution that works for all values of $J$. However, $J$ can be arbitrary and not smooth, which makes designing a general law impossible. Instead, we follow Rybkin et al. [39] and use the notion of a total budget $\mathcal{F} = \mathcal{C} + \delta \cdot \mathcal{D}$ as a substitute for $J$. Similarly to $J$, the budget $\mathcal{F}$ increases as the complexity of policy learning increases.

That is, for a well-behaved TD-learning algorithm with the "optimal" hyperparameters, $J$ will be some unknown monotonic function of $\mathcal{F}$. Using this intuition, we will now demonstrate a solution to compute allocation that optimizes $\mathcal{F}$, therefore also optimizing $J$. Similarly, we will be able to extrapolate our solution to higher $\mathcal{F}$, and thus higher $J$.

We produce a solution to Problem 4.1, part 2, by observing that $\mathcal{C}$ and $\mathcal{D}$ evolve predictably as a function of $\mathcal{F}$, in line with previous work [39]:

$$\mathcal{C}^*(\mathcal{F}_0) = \left(\frac{a_\mathcal{C}}{\mathcal{F}_0}\right)^{\alpha_\mathcal{C}}, \quad \mathcal{D}^*(\mathcal{F}_0) = \left(\frac{b_\mathcal{C}}{\mathcal{F}_0}\right)^{\beta_\mathcal{C}}. \tag{7.3}$$

Figure 7: **Optimal data $\mathcal{D}(\mathcal{F}_0)$ and compute $\mathcal{C}(\mathcal{F}_0)$ fits for a given budget $\mathcal{F}_0$.** Return $J$ is denoted in color, showing how increased budgets correspond to higher returns. Similar to [39], we are able to allocate resources across data and compute in a predictable way, while accounting for the effect of both model size and UTD.

**Evaluation.** We show that this dependency is predictable in Figure 7, including evaluating confidence interval and extrapolation to higher budgets for this fit. This allows us to optimally allocate resources for higher values of budget or return across data and compute.

> **Scaling Observation 3: Optimal partitioning between data and compute**
>
> Optimal scaling for data $\mathcal{C}$ and compute $\mathcal{D}$ are predictable functions of the total budget $\mathcal{F}_0$, as a power law in Eq. (7.3).

Now, we extend this analysis to allocating compute across UTD and model size as a function of the budget. We use the same power law form:

$$\sigma_{\mathcal{F}}^*(\mathcal{F}_0) = \left(\frac{a_{\mathcal{F}}}{\mathcal{F}_0}\right)^{\alpha_{\mathcal{F}}}, \quad N_{\mathcal{F}}^*(\mathcal{F}_0) = \left(\frac{b_{\mathcal{F}}}{\mathcal{F}_0}\right)^{\beta_{\mathcal{F}}}. \tag{7.4}$$

> **Scaling Observation 4: Optimal partitioning of budget between UTD and model size**
>
> Optimal scaling for UTD $\sigma$ and model size $N$ depends as a power law on the budget $\mathcal{F}$, as in Eq. (7.3). We can estimate the optimal allocation trend using this power law, and estimate robustness of perfomance to allocation as the variance of this trend.

**Implications.** We show results for two challenging tasks in Figure 8 and further results in Appendix D. We observe the coefficients $\alpha_{\mathcal{F}}, \beta_{\mathcal{F}}$ for resource allocation vary between tasks, showing that for some tasks scaling model size or UTD is more or less important. Further, we observe that different tasks vary in the amount of variance, seen as the size of the confidence interval in Figure 8. This shows that for some tasks, precisely setting model size and UTD is important; while other tasks allow to trade off model size and UTD without a big decrease in performance. Our experimental procedure enables practitioners to make these workflow decisions based on the relationships that we fit in this paper.

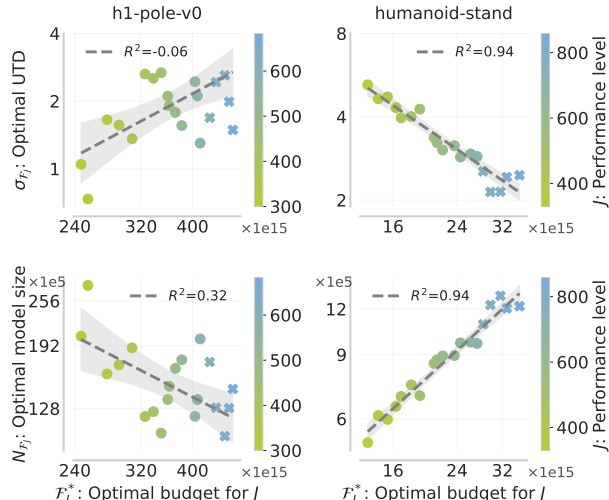

Figure 8: **Optimal UTD** $\sigma(\mathcal{F}_0)$ **and model size** $N(\mathcal{F}_0)$, with extrapolation to higher budgets or returns. While for some tasks it is necessary to set values precisely, other tasks allow some variation in model size and UTD as indicated by variance.

## 8 Discussion

We have established scaling laws for value-based RL allowing compute scaling in an optimal manner. Specifically, we provide a way to scale batch size, UTD, model size, as well as data budget, and provide scaling laws that estimate tradeoffs between these quantities. These laws are informed by our novel analysis of the impact of scaling on overfitting in TD-learning. We also saw that in some environments several configurations of the hyperparameters we studied could broadly be considered compute-optimal, which reflected as a benign relative error in our fits. We were limited in how many variables we can study due to the necessity of running higher-dimensional grid searches for every new variable. Building on our results, future work will study other important hyperparameters, such as learning rate and the critic update ratio. Further, while our work is limited to challenging simulated robotic tasks, future work will study large scale domains such as visual and language domains using larger scale models. The analysis and the laws presented in this work are a step towards training TD-learning methods at a scale similar to other modern machine learning approaches.

## Acknowledgements

We would like to thank Amrith Setlur, Seohong Park, Colin Li, and Mitsuhiko Nakamoto for feedback on an earlier version of this paper. We thank the TRC program at Google Cloud for providing TPU sources that supported this work. We thank NCSA Delta cluster for providing GPU resources that supported the experiments in this work. This research was supported by ONR under N00014-24-12206, N00014-22-1-2773, and ONR DURIP grant, with compute support from the Berkeley Research Compute, Polish high-performance computing infrastructure, PLGrid (HPC Center: ACK Cyfronet AGH), that provided computational resources and support under grant no. PLG/2024/017817. Pieter Abbeel holds concurrent appointments as a Professor at UC Berkeley and as an Amazon Scholar. This work was done at UC Berkeley and CMU, and is not associated with Amazon.

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

# A  Details on Deriving Scaling Fits

**FLOPs calculation.** We inherit the definition from Rybkin et al. [39], so that

$$\mathcal{C}(\sigma, N) = k \, \sigma \, N \, \mathcal{D}(\sigma, N) \tag{A.1}$$

for a constant $k$ not dependent on $\sigma$ and $N$. We follow the standard practice of updating the critic, target, and actor all $\sigma$ times for each new data point collected (Algorithm 1).

## A.1  Maximal compute efficiency for data $\leq \mathcal{D}_0$

As described in Section 7, the number of data points needed to achieve performance $J$ is equal to

$$\mathcal{D}_J(\sigma, N) \approx \mathcal{D}_J^{\min} + \left( \frac{a_J}{\sigma} \right)^{\alpha_J} + \left( \frac{b_J}{N} \right)^{\beta_J}, \tag{A.2}$$

where $\mathcal{D}_J^{\min}, a_J, \alpha_J, b_J, \beta_J > 0$ are constants not dependent on $\sigma$ and $N$. We first present a closed-form solution to the simpler optimization problem in Eq. (A.3). This will enable us to characterize the solution to Problem 4.1, part 1, which does not have a closed-form solution in terms of $\mathcal{C}_0$ but can be easily estimated.

**Proposition A.1.** If $\alpha_J < 1$ or $\beta_J < 1$, there exists a unique optimum

$$(\sigma^*, N^*) := \arg \min_{(\sigma, N)} \ \mathcal{C}_J(\sigma, N) \ \text{ s.t. } \ \mathcal{D}_J(\sigma, N) \leq \mathcal{D}_0. \tag{A.3}$$

Moreover,

$$\sigma^* = a_J \left( \frac{1 + \frac{\alpha_J}{\beta_J}}{\mathcal{D}_0 - \mathcal{D}^{\min}} \right)^{1/\alpha_J} \qquad N^* = b_J \left( \frac{1 + \frac{\beta_J}{\alpha_J}}{\mathcal{D}_0 - \mathcal{D}^{\min}} \right)^{1/\beta_J} \tag{A.4}$$

satisfy the following relation:

$$N^* = \left( \frac{\beta_J b^{\beta_J}}{\alpha_J a^{\alpha_J}} \right)^{1/\beta_J} (\sigma^*)^{\alpha_J / \beta_J}. \tag{A.5}$$

*Proof.* For ease of notation, we drop the subscript $J$ throughout this derivation. Since there exist sufficiently large $(\sigma, N)$ for which $\mathcal{D}(\sigma, N) < \mathcal{D}_0$, Slater's conditions are satisfied, and the KKT conditions are necessary and sufficient for optimality. Let

$$\mathcal{L}(\sigma, N) = k \, \sigma \, N \, \mathcal{D}(\sigma, N) + \lambda(\mathcal{D}(\sigma, N) - \mathcal{D}_0) \tag{A.6}$$

denote the Lagrangian. Now, we will solve for $(\tilde{\sigma}, \tilde{N}, \tilde{\lambda})$ satisfying the KKT conditions. The stationarity conditions are

$$\frac{\partial \mathcal{L}}{\partial \sigma} = 0 \implies (k \, \tilde{\sigma} \, \tilde{N} + \tilde{\lambda}) \, \alpha \, a^\alpha \, \tilde{\sigma}^{-\alpha - 1} = k \tilde{N} \mathcal{D}(\tilde{\sigma}, \tilde{N}) \tag{A.7}$$

$$\frac{\partial \mathcal{L}}{\partial N} = 0 \implies (k \, \tilde{\sigma} \, \tilde{N} + \tilde{\lambda}) \, \beta \, b^\beta \, \tilde{N}^{-\beta - 1} = k \tilde{\sigma} \mathcal{D}(\tilde{\sigma}, \tilde{N}). \tag{A.8}$$

Complementary slackness implies that

$$\tilde{\lambda}(\mathcal{D}(\tilde{\sigma}, \tilde{N}) - \mathcal{D}_0) = 0. \tag{A.9}$$

We claim that $\tilde{\lambda} > 0$. Assume for the sake of contradiction that $\tilde{\lambda} = 0$. Substituting into Equations (A.7) and (A.8), we obtain

$$\alpha \, a^\alpha \, \tilde{\sigma}^{-\alpha} = \beta \, b^\beta \, \tilde{N}^{-\beta} = \mathcal{D}(\tilde{\sigma}, \tilde{N}) > \max \left\{ a^\alpha \tilde{\sigma}^{-\alpha}, b^\beta \tilde{N}^{-\beta} \right\}. \tag{A.10}$$

But the last inequality contradicts $\alpha < 1$ or $\beta < 1$, concluding the claim.

It follows that $\mathcal{D}(\tilde{\sigma}, \tilde{N}) = \mathcal{D}_0$. Dividing Eq. (A.7) by Eq. (A.8), we obtain

$$\frac{\alpha \, a^\alpha \, \tilde{\sigma}^{-\alpha-1}}{\beta \, b^\beta \, \tilde{N}^{-\beta-1}} = \frac{\tilde{N}}{\tilde{\sigma}}, \tag{A.11}$$

or equivalently

$$\tilde{N} = \left(\frac{\beta b^\beta}{\alpha a^\alpha}\right)^{1/\beta} \sigma^{\alpha/\beta}. \tag{A.12}$$

Substituting into the active constraint $\mathcal{D}(\tilde{\sigma}, \tilde{N}) = \mathcal{D}_0$, we obtain

$$\tilde{\sigma} = a \left(\frac{1 + \frac{\alpha}{\beta}}{\mathcal{D}_0 - \mathcal{D}^{\min}}\right)^{1/\alpha}, \quad \tilde{N} = b \left(\frac{1 + \frac{\beta}{\alpha}}{\mathcal{D}_0 - \mathcal{D}^{\min}}\right)^{1/\beta}. \tag{A.13}$$

Thus, $(\tilde{\sigma}, \tilde{N})$ is the unique KKT solution, and thus the unique optima. $\qquad\square$

## A.2   Maximal data efficiency for compute $\leq \mathcal{C}_0$

We are now equipped to solve the optimization problem presented in Problem 4.1, part 1. Although we cannot solve for the optimal $(\sigma^*, N^*)$ directly, the following proposition shows that the set of optimal solutions obtained by varying the compute budget $\mathcal{C}_0$ matches exactly the set of solutions obtained by varying the data budget $\mathcal{D}_0$ in Proposition A.1. This equivalence reduces the original problem to a simpler surrogate. Using Eq. (A.5), it is straightforward to compute the optimum numerically.

**Proposition A.2.** Suppose $\alpha_J < 1$ or $\beta_J < 1$, and assume the data and compute formulations established in Equations (A.1) and (A.2). Let

$$\mathcal{C}_J^{\min} := \min_{\sigma > 0, N > 0} \mathcal{C}_J(\sigma, N). \tag{A.14}$$

For a fixed budget $\mathcal{C}_0 \geq \mathcal{C}_J^{\min}$, write

$$(P_1) \qquad \mathcal{D}^* = \min_{(\sigma, N)} \ \mathcal{D}_J(\sigma, N) \ \text{ s.t. } \ \mathcal{C}_J(\sigma, N) \leq \mathcal{C}_0 \tag{A.15}$$

and

$$(P_2) \qquad \min_{(\sigma, N)} \ \mathcal{C}_J(\sigma, N) \ \text{ s.t. } \ \mathcal{D}_J(\sigma, N) \leq \mathcal{D}^*. \tag{A.16}$$

Each problem admits a unique solution $(\sigma^*, N^*)$, and these solutions coincide.

*Proof.* As before, we drop the subscript $J$.

We first justify the existence of a global minimizer to $(P_1)$ over $(0, \infty)^2$. As $\sigma \to 0^+$ or $N \to 0^+$, then $(a/\sigma)^\alpha \to \infty$ or $(b/N)^\beta \to \infty$, hence $D(\sigma, N) \to \infty$. If $\sigma, N \to \infty$, then $\mathcal{C}(\sigma, N) \geq k\sigma N \mathcal{D}^{\min} \to \infty$, contradicting $\mathcal{C} \leq \mathcal{C}_0$. Thus, the feasible set $\{\mathcal{C} \leq \mathcal{C}_0\}$ is coercive, and by continuity of $\mathcal{C}$ and $\mathcal{D}$, $(P_1)$ attains a global minimizer.

Proposition A.1 shows that there exists $\lambda > 0$ such that the KKT conditions for $(P_1)$ hold,

$$\frac{\partial \mathcal{D}}{\partial \sigma}(\sigma^*, N^*) + \lambda \frac{\partial \mathcal{C}}{\partial \sigma}(\sigma^*, N^*) = 0 \tag{A.17}$$

$$\frac{\partial \mathcal{D}}{\partial N}(\sigma^*, N^*) + \lambda \frac{\partial \mathcal{C}}{\partial N}(\sigma^*, N^*) = 0 \tag{A.18}$$

$$\mathcal{C}(\sigma^*, N^*) = \mathcal{C}_0. \tag{A.19}$$

For $(P_2)$, the KKT conditions imply that there exists $\mu \geq 0$ such that

$$\frac{\partial \mathcal{C}}{\partial \sigma}(\sigma^\dagger, N^\dagger) + \mu \frac{\partial \mathcal{D}}{\partial \sigma}(\sigma^\dagger, N^\dagger) = 0 \tag{A.20}$$

$$\frac{\partial \mathcal{C}}{\partial N}(\sigma^\dagger, N^\dagger) + \mu \frac{\partial \mathcal{D}}{\partial N}(\sigma^\dagger, N^\dagger) = 0 \tag{A.21}$$

$$\mathcal{D}(\sigma^\dagger, N^\dagger) = \mathcal{D}^*. \tag{A.22}$$

If $\alpha < 1$, then

$$\frac{\partial \mathcal{D}}{\partial \sigma} = -\alpha a^\alpha \sigma^{-\alpha-1} < 0 \tag{A.23}$$

$$\frac{\partial \mathcal{C}}{\partial \sigma} = -k\sigma N \alpha a^\alpha \sigma^{-\alpha-1} + kN(\mathcal{D}_{\min} + a^\alpha \sigma^{-\alpha} + b^\beta N^{-\beta}) > 0, \tag{A.24}$$

so $\mu > 0$. In the other case, if $\beta < 1$, then $\frac{\partial \mathcal{D}}{\partial N} < 0$ and $\frac{\partial \mathcal{C}}{\partial N} > 0$, so $\mu > 0$.

Since the first solution is additionally given to satisfy $\mathcal{D}(\sigma^*, N^*) = \mathcal{D}^*$, these systems are identical, and so must be their solutions, $(\sigma^*, N^*, \lambda) = (\sigma^\dagger, N^\dagger, 1/\mu)$. Uniqueness in proposition A.1 implies uniqueness in $(P_1)$. $\qquad\square$

### A.3 Maximal performance for budget $\leq \mathcal{F}_0$

Performance level $J$ is task-dependent and is not guaranteed to satisfy any general properties, so modeling part 2 of Problem 4.1 directly is impossible. However, given a particular value of $J$, we can compute the UTD ratio $\sigma_{\mathcal{F}_J}$ and model size $N_{\mathcal{F}_J}$ that uniquely minimize the total budget $\mathcal{F}_J(\sigma, N) = \mathcal{C}_J(\sigma, N) + \delta \cdot \mathcal{D}_J(\sigma, N)$ (see Proposition A.3). We run this procedure for $J_1, \ldots, J_m \in [J_{\min}, J_{\max}]$, as described in Appendix B.

We expect that a higher budget will ultimately yield higher performance under the best hyperparameter configuration. This procedure yields several points $\{(J_i, \mathcal{F}_{J_i})\}_{i=1}^m$ along the Pareto frontier $J \mapsto \min_{\sigma, N} \mathcal{F}_J(\sigma, N)$, as shown in Figure 9. Importantly, we do not directly model this curve, and only need its existence, continuity, and monotonically increasing nature for our fits. Consequently, its inverse is continuous and monotonically increasing. Therefore, for a given budget $\mathcal{F}_{J_i}$, $1 \leq i \leq m$, the performance level $J_i$ is optimal for that budget, i.e.

$$(\sigma_{\mathcal{F}}^*(\mathcal{F}_{J_i}), N_{\mathcal{F}}^*(\mathcal{F}_{J_i})) = \arg\max_{(\sigma, N)} \; J(\pi_{\mathrm{Alg}}(\sigma, N)) \quad \text{s.t.} \quad \mathcal{C} + \delta\mathcal{D} \leq \mathcal{F}_{J_i}.$$

This procedure yields $m$ points along the solution to Problem 4.1, part 2. Since data efficiency is predictable, we can therefore constrain the budget to model $\sigma_{\mathcal{F}}^*$, $N_{\mathcal{F}}^*$ as in Eq. (7.4).

**Proposition A.3.** Suppose $(\alpha_J, \beta_J) \in (0, 1)$, and fix $\delta > 0$. Consider the unconstrained minimization $\min_{\sigma, N} \mathcal{F}_J(\sigma, N)$. The optimum $(\sigma^*, N^*)$ is unique and satisfies Eq. (A.5).

*Proof.* As either $\sigma \to 0^+$ or $N \to 0^+$, the term $\delta\mathcal{D} \to \infty$. As $\sigma, N \to \infty$, the term $\mathcal{C} \geq k\sigma N\mathcal{D}^{\min} \to \infty$. By the same logic as the proof of Proposition A.2, a global minimizer exists.

Then, the objective is exactly the same as Eq. (A.6), with $\lambda$ replaced by $\delta$, and the $\mathcal{D}_0$ constant offset removed. Thus, the same logic in the proof of Proposition A.1 applies, and we obtain the same relation Eq. (A.5).

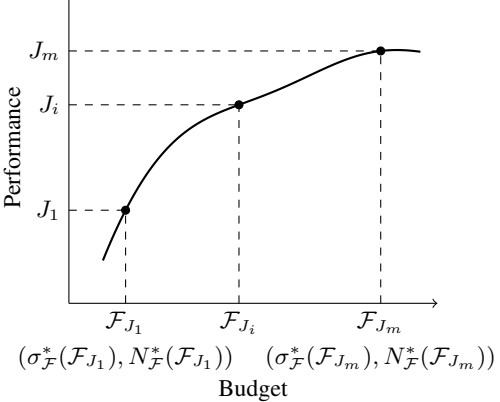

Figure 9: A (hypothetical) depiction of the performance–budget Pareto frontier we implicitly model. For each $J_i$, we compute the budget-minimizing UTD ratio $\sigma_{\mathcal{F}}^*(\mathcal{F}_{J_i})$ and model size $N_{\mathcal{F}}^*(\mathcal{F}_{J_i})$. We can then discard the $y$-axis, leaving us with a relationship between budget $\mathcal{F}$ and $(\sigma_{\mathcal{F}}^*, N_{\mathcal{F}}^*)$.

## B Experiment Details

For our experiments, we use a total of 17 tasks from two benchmarks (DeepMind Control [51] and HumanoidBench [42]), listed in Table 2, with the BRO algorithm and architecture [34]. We additionally use 2 tasks from HumanoidBench (`h1-crawl`, `h1-stand`) with SimbaV2 [26]. As described in Appendix C, we normalize our returns to $[0, 1000]$; optimal $\pi$ returns are pre-normalized. For HumanoidBench, we report the returns listed by authors as the "success bar," even though it is possible to achieve a higher return. Our experiments fit $\mathcal{D}_J(\sigma, N)$ for 20 normalized performance thresholds $J$, spaced uniformly between $J_{\min}$ and $J_{\max}$, inclusive; 20 is an arbitrary choice that we made so as to obtain useful insights about our method while not overwhelming the reader.

Table 2: Tasks used in presented experiments.

| Domain | Task | Optimal $\pi$ Returns | $J_{\min}$ | $J_{\max}$ | $\delta$ |
|---|---|---|---|---|---|
| HumanoidBench | `h1-crawl` | 700 | 450 | 780 | 2e12 |
| | `h1-pole` | 700 | 300 | 680 | 5e11 |
| | `h1-stand` | 800 | 200 | 660 | 5e11 |
| DMC | `humanoid-stand` | 1000 | 300 | 850 | 5e10 |
| DMC-Medium | `acrobot-swingup` | 1000 | 150 | 400 | 1e11 |
| | `cheetah-run` | 1000 | 400 | 750 | 1e11 |
| | `finger-turn-hard` | 1000 | 400 | 900 | 1e11 |
| | `fish-swim` | 1000 | 200 | 710 | 1e11 |
| | `hopper-hop` | 1000 | 150 | 320 | 1e11 |
| | `quadruped-run` | 1000 | 200 | 790 | 1e11 |
| | `walker-run` | 1000 | 350 | 730 | 1e11 |
| DMC-Hard | `dog-run` | 1000 | 100 | 270 | 1e11 |
| | `dog-trot` | 1000 | 100 | 580 | 1e11 |
| | `dog-stand` | 1000 | 100 | 910 | 1e11 |
| | `dog-walk` | 1000 | 100 | 860 | 1e11 |
| | `humanoid-run` | 1000 | 75 | 190 | 1e11 |
| | `humanoid-walk` | 1000 | 200 | 650 | 1e11 |

Table 3: Configurations for ORIGINAL 3-dimensional grid searches.

| Task | UTD ratio $\sigma$ | Critic width | Possible batch sizes |
|---|---|---|---|
| `h1-crawl` | 1, 2, 4, 8 | 256, 512, 1024, 2048 | 16, 32, 64, 128, 256, 512, 1024, 2048 |
| `h1-pole` | 1, 2, 4, 8 | 256, 512, 1024, 2048 | 64, 128, 256, 512, 1024, 2048 |
| `h1-stand` | 1, 2, 4, 8 | 256, 512, 1024, 2048 | 128, 256, 512, 1024, 2048, 4096 |
| `humanoid-stand` | 1, 2, 4, 8 | 128, 256, 512, 1024, 2048 | 64, 128, 256, 512, 1024 |

## B.1 Hyperparameter Sweep Details

Out of the 17 tasks, we run a full 3-dimensional grid search $B \times N \times \sigma$ on 4 of them: 3 tasks from HumanoidBench and Humanoid-Stand from DMC. Due to the computational requirements of running a large grid search for obtaining the scaling fits, we use a constant network depth (2 BroNet blocks [34]) and learning rate (3e-4) throughout our experiments and run at least 5 random seeds per configuration. From these experiments, we follow the procedure described in Appendix C to estimate a batch size rule (Figure 11). A superset of the configurations run in this three-dimensional grid search are listed as ORIGINAL in Table 3. Out of the listed batch sizes in Table 3, we run at least 4 consecutive values of batch sizes for each $(\sigma, N)$, such that the empirically most performant batch size is neither the minimum nor maximum of the range. Since a full 3D-sweep is expensive, this heuristic enables us to effectively reduce the total number of experiments we need to run to estimate batch size fits. For instance, for small model sizes and low UTD values on `h1-crawl`, this amounts to simply running batch sizes up to 64, since performance decreases monotonically as the batch size increases.

Based on these runs, we set $J_{\min}$ and $J_{\max}$, as described in the following subsection. This enables us to establish a batch size rule (Eq. (6.1)), where the "best" batch size uses the least amount of data to achieve performance $J_{\max}$. To evaluate our batch size rule $B^*(\sigma, N)$, we run a 2-dimensional sweep using our proposed batch sizes on INTERPOLATED and EXTRAPOLATED UTD ratios $\sigma$ and model sizes $N$. The configurations are listed in Table 4. Note that we did not study extrapolation of model size on `humanoid-stand`, since we already noticed that a width of 2048 performed worse than a model width of 1024 at low UTD values.

Using various combinations of these measurements, we can fit data efficiency (Eq. (7.1)). InSection 7.1, we evaluate the absolute relative error of the fit prediction with respect to the ground truth data efficiency on each of the datasets, when the fit solely uses ORIGINAL data and is evaluated on ORIGINAL, INTERPOLATED, and EXTRAPOLATED data. Our final $\mathcal{D}$, as described elsewhere in the paper, is fitted on all three datasets, ORIGINAL, INTERPOLATED, and EXTRAPOLATED.

Table 4: Configurations for INTERPOLATED and EXTRAPOLATED.

| Dataset | Task | UTD ratio $\sigma$ | Critic width |
|---|---|---|---|
| INTERPOLATED | h1-crawl | 3, 6, 12 | 368, 720, 1456 |
| | h1-pole | 3, 6, 12 | 368, 720, 1456 |
| | h1-stand | 3, 6, 12 | 368, 720, 1456 |
| | humanoid-stand | 3, 6, 12 | 176, 368, 720, 1456 |
| $N$ EXTRAPOLATED | h1-crawl | 1, 2, 4, 8, 16 | 4096 |
| | h1-pole | 1, 2, 4, 8, 16 | 4096 |
| | h1-stand | 1, 2, 4, 8, 16 | 4096 |
| $\sigma$ EXTRAPOLATED | h1-crawl | 16 | 256, 512, 1024, 2048 |
| | h1-pole | 16 | 256, 512, 1024, 2048 |
| | h1-stand | 16 | 256, 512, 1024, 2048 |
| | humanoid-stand | 16 | 128, 256, 512, 1024, 2048 |

The other 13 tasks are from DMC, which we group as DMC-medium and DMC-hard following Nauman et al. [34]. For obtaining these fits, we borrow the data directly from Nauman et al. [34]: the authors of this prior work ran 10 random seeds at a constant batch size 128 and learning rate 3e-4 on several UTD (1, 2, 5, 10, 15) and model size (Table 7 in [34]) configurations. Due to the lack of appropriately set batch size in these experiments borrowed from prior work, the data does not accurately represent the best achievable data efficiency, and in some cases increasing UTD or model size worsens performance. In these cases, fitting $\mathcal{D}$ per task can result in instability, where the exponents $\alpha_J$, $\beta_J$ are driven to 0. To counteract this, we use two approaches:

1. **Share parameters $\alpha_J$, $\beta_J$ of the fit over tasks** as follows:

$$\mathcal{D}_J^{\text{env}}(\sigma, N) \approx \mathcal{D}_J^{\text{env min}} + \left(\frac{a_J^{\text{env}}}{\sigma}\right)^{\alpha_J} + \left(\frac{b_J^{\text{env}}}{N}\right)^{\beta_J}. \quad \text{(B.1)}$$

   Conceptually, this forces the slope of the compute-optimal line prescribed by Eq. (A.5) to be shared across tasks within the same domain, but allows for a different intercept. This results in variance reduction in the fitting procedure.

2. **Average over multiple tasks according to the procedure in Appendix C.** We present these fits in the main paper to improve clarity and reduce clutter (Figure 6). This method essentially treats the benchmark as a single task and fits an average amount of data required to achieve some performance.

**Selecting experimental constants.** To select $J_{\max}$, we first group by the UTD ratio $\sigma$ and model size $N$. Out of each group, we select the run with the highest final Monte-Carlo returns (over all batch sizes). Over these runs, we set $J_{\max}$ as the highest return threshold that 80% of the runs reach.

We heuristically select $J_{\min}$ as the lowest return threshold such that configurations that eventually reach performance $J_{\max}$ "look sufficiently different," i.e. there are configurations with batch sizes $B_1$, $B_2$ such that their confidence intervals $[\mathcal{D}_{J_{\min}} - \sigma_{J_{\min}}, \mathcal{D}_{J_{\min}} + \sigma_{J_{\min}}]$ do not overlap. Here $\mathcal{D}$ denotes the true (not fitted) amount of data required to reach the performance level, and $\sigma$ is the standard deviation given by the procedure described in Appendix C.

We select $\delta$ in the budget formula $\mathcal{F} = \mathcal{C} + \delta \mathcal{D}$ so that $\delta \mathcal{D}$ represents the real-time cost of environment steps, as measured in FLOPs. Our procedure is as follows:

1. Pick the run that achieves performance $J_{\max}$ within the lowest wall-clock time.

2. Based on timing statistics from this run, set

$$\delta \approx \frac{\text{FLOPs/grad steps} \times \text{grad steps/sec}}{\text{env steps/sec}}. \quad \text{(B.2)}$$

The resulting expression for $\mathcal{F}$ is therefore a proxy for wall clock time.

---
**Algorithm 1** Training loop drop-ins for any value-based algorithm
---
1: Initialize environment $p$
2: Initialize replay buffer $\mathcal{P}$
3: Initialize parameter vectors $\theta$ (critic), $\bar{\theta}$ (target critic), $\phi$ (actor)
4: Initialize validation environment $p^{\text{val}}$
5: Initialize validation replay buffer $\mathcal{P}^{\text{val}}$          *// size $|\mathcal{P}|/k$*
6: Initialize passive critic parameter vector $\theta^{\text{passive}}$      *// possibly different size than $\theta$*
7: **for** each iteration **do**
8:      **for** each environment step **do**
9:          $a_t \sim \pi_\phi(a_t|s_t)$
10:         $s_{t+1} \sim p(s_{t+1}|s_t, a_t)$
11:         $\mathcal{P} \leftarrow \mathcal{P} \cup \{(s_t, a_t, r(s_t, a_t), s_{t+1})\}$
12:         **if** $t \bmod k = 0$ **then**         *// do validation less frequently to avoid overhead*
13:            $a_t^{\text{val}} \sim \pi_\phi(a_t^{\text{val}}|s_t^{\text{val}})$
14:            $s_{t+1}^{\text{val}} \sim p^{\text{val}}(s_{t+1}^{\text{val}}|s_t^{\text{val}}, a_t^{\text{val}})$
15:            $\mathcal{P}^{\text{val}} \leftarrow \mathcal{P}^{\text{val}} \cup \{(s_t^{\text{val}}, a_t^{\text{val}}, r(s_t^{\text{val}}, a_t^{\text{val}}), s_{t+1}^{\text{val}})\}$
16:         **end if**
17:      **end for**
18:      **for** each update **do**
19:         Sample training batch $x \sim \mathcal{P}$
20:         **for** $\sigma$ gradient steps **do**
21:            $\theta \leftarrow \theta - \eta_{\text{critic}} \nabla_\theta \mathcal{L}_{\text{critic}}(x; \theta, \bar{\theta})$
22:            $\theta^{\text{passive}} \leftarrow \theta^{\text{passive}} - \eta_{\text{critic}} \nabla_{\theta^{\text{passive}}} \mathcal{L}_{\text{critic}}(x; \theta^{\text{passive}}, \bar{\theta})$
23:            $\phi \leftarrow \phi - \eta_{\text{actor}} \nabla_\phi \mathcal{L}_{\text{actor}}(x; \theta, \phi)$
24:            $\bar{\theta} \leftarrow \tau\theta + (1 - \tau)\bar{\theta}$
25:         **end for**
26:         **if** logging **then**
27:            Sample validation batch $x^{\text{val}} \sim \mathcal{P}^{\text{val}}$
28:            $\mathcal{L}_{\text{critic}}^{\text{val}} \leftarrow \mathcal{L}_{\text{critic}}(x^{\text{val}}; \theta, \bar{\theta})$
29:            $\mathcal{L}_{\text{critic}}^{\text{passive}} \leftarrow \mathcal{L}_{\text{critic}}(x; \theta^{\text{passive}}, \bar{\theta})$
30:         **end if**
31:      **end for**
32: **end for**
---

## B.2    Detailed Explanations for How to Obtain Main Paper Figures

**Figure 2.** Standard off-policy online RL trains on data sampled from a replay buffer, which is regularly augmented with data from the environment. We construct a held-out dataset of transitions following the same distribution as the training replay buffer. To do so, we create a validation environment, which is identical to the training environment with a different random seed, and a corresponding validation replay buffer. This allows us to measure the validation TD-error, i.e. the TD-error of the critic against the target on data sampled from the validation replay buffer. Algorithmic details are described in Algorithm 1 in blue.

**Figure 4.** The passive critic regresses onto the target produced by the main critic, and is trained using a similar procedure as the main critic. We report the TD-error of the passive critic against the TD-target on validation data. Algorithmic details are described in Algorithm 1 in green.

**Figure 5.** We describe our batch size fitting procedure in Appendix C.

**Figure 6.** Circles represent the true data efficiencies on our ORIGINAL UTD ratios and model sizes. Using this data, we fit a batch size rule $B^*(\sigma, N)$ (Eq. (6.1)), and run experiments using our batch size rule on INTERPOLATED and EXTRAPOLATED UTD ratios and model sizes. Then, we fit data efficiency $\mathcal{D}_{J_{\max}}(\sigma, N)$ (Eq. (7.1)) on all of the data, where $J_{\max}$ is listed in Table 2. The iso-data contours are predictions from the fit, and the log-log-line containing compute-optimal points follows the formula in Eq. (7.2).

**Figure 7.** We fit $\mathcal{D}_{J_i}$ independently for each $J_i$. Following Appendix A.3, we numerically solve for the optimum $(\sigma_{\mathcal{F}}^*(\mathcal{F}_{J_i}), N_{\mathcal{F}}^*(\mathcal{F}_{J_i}))$. We plot $\mathcal{D}$ and $\mathcal{C}$ for these optima against $\mathcal{F}_{J_i}$. Out of these $m = 20$ points, we fit a line to the bottom 15 of them and mark the top 5 as budget extrapolation. We record $R^2$ between the log-linear fit and log-$y$ values over all 20 points.

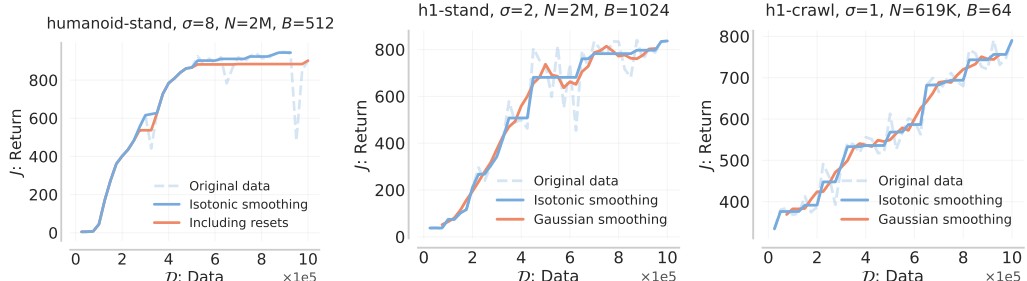

Figure 10: A demonstration of our MC returns preprocessing. **Left:** Full-parameter resets introduce variance in returns; we remove the dips before running isotonic regression. **Middle:** Gaussian smoothing can lead to under-smoothing the returns, making data efficiency more difficult to fit. **Right:** Gaussian smoothing can lead to over-smoothing the returns, e.g. at 625K env steps, Gaussian-smoothed returns are higher than the maximum returns achieved up to that point.

**Figure 8.** Same method as Figure 7.

## C  Additional Details on the Fitting Procedure

**Preprocessing return values.** Our fits require estimates of the data and compute needed by a given run to reach a target performance level. The BRO algorithm [34] employs full-parameter resets as a form of plasticity regularization [35], reinitializing the agent every 2.5M gradient steps to encourage better exploration and long-term learning. However, these resets induce abrupt drops in Monte Carlo (MC) returns, which do not reflect a true degradation in learning quality. Instead, returns typically recover quickly and often surpass pre-reset levels. Including these transient dips in the MC curve would artificially inflate the estimated data and compute required to reach a given performance threshold. To obtain a cleaner, more consistent signal of learning progress, we therefore remove post-reset return drops from our analysis (Figure 10, left). This allows us to more accurately model the intrinsic data efficiency of the algorithm, independent of reset-induced variance.

Following [39], we then process the return values with isotonic regression [1], which transforms the return values to the most aligned nondecreasing sequence of values that can then be used to estimate $\mathcal{D}_J$ (Figure 10, middle and right). This procedure enables us to fit the minimum number of samples needed to reach a given performance level, regardless of whether the performance drops later in training. It also reduces variance compared to the naive approach of measuring the data efficiency directly on each random seed.

**Uncertainty-adjusted optimal batch size.** We follow Rybkin et al. [39] to compute uncertainty-adjusted optimal batch sizes, since the precision of the fit $B$ would otherwise be limited by the granularity of our grid search. We run $K = 100$ bootstrap estimates by sampling $n$ random seeds with replacement out of the original $n$ random seeds, applying isotonic regression, and selecting the optimal batch size $B_k$ by data efficiency to threshold $J_{\max}$. Since these batch sizes can span multiple orders of magnitude (Table 3), we report the mean of these bootstrapped estimates in log space as the "best" batch size:

$$B_{\text{bootstrap}} = \exp\left(\frac{1}{K}\sum_{k=1}^{K}\log B_k\right). \tag{C.1}$$

Additionally, considering the set of bootstrapped data efficiencies to reach a given performance threshold $J$, this procedure also yields an estimate of the standard deviation of the data efficiency.

**Fitting procedure.** Prior work fits the data using a brute force grid search followed by LBFG-S [16, 39]. Empirically, we found that the quality of the resulting fit is highly dependent on the initial point found by brute force, and the bounds of the brute force grid must be tuned per fit. To resolve these issues, we use the following procedure:

1. Normalize the inputs $x$ to $[\ell, h] = [0.5, 2]$ in log space via

$$s = \frac{\log(\max x) - \log(\min x)}{\log h - \log \ell} \tag{C.2}$$

$$m = \log(\min x) - s \log \ell \tag{C.3}$$

$$x' = \exp\left(\frac{\log x - m}{s}\right), \tag{C.4}$$

and normalize the output $y$ by dividing by the mean, $y' = y/\overline{y}$. This results in a more numerically stable fitting procedure, since $\sigma \in [1, 20]$ and $N \in [1e5, 2e8]$ are otherwise on very different scales.

2. Define $\theta' = \mathrm{softplus}(\theta) = \log(1 + \exp(\theta))$ for all "raw" parameters $\theta \in \mathbb{R}$. Softplus is a smooth approximation to ReLU and forces fit parameters to be positive, and empirically tends to improve fitting stability. For example, to fit data efficiency, we optimize over $[\theta_{\mathcal{D}^{\min}}, \theta_a, \theta_b, \theta_\alpha, \theta_\beta] \in \mathbb{R}^5$, and extract e.g. $\mathcal{D}^{\min} = \mathrm{softplus}(\theta_{\mathcal{D}^{\min}})$.

3. Use LBFG-S to optimize over raw parameters. We use MSE in log space as the objective: $\mathcal{L}(y, \hat{y}) = (\log y - \log \hat{y})^2$.

4. Apply softplus and correct the parameters for normalization.

Empirically, we find that initializing all raw parameters as zero generally works well.

**Aggregate data efficiency.** In Figure 6, we show data efficiency fits aggregated over multiple tasks. We follow Rybkin et al. [39]: first, normalize the data efficiency $\mathcal{D}_J^{\mathrm{env}}$ by intra-environment medians $\mathcal{D}_J^{\mathrm{env\ med}} = \mathrm{median}\left\{\mathcal{D}_J^{\mathrm{env}}(\sigma, N)\right\}_{\sigma, N}$. To interpret the normalized data efficiency on the same scale as the original data, we write $\mathcal{D}_J^{\mathrm{med}} = \mathrm{median}\left\{\mathcal{D}_J^{\mathrm{env\ med}}\right\}_{\mathrm{env}}$, so that $\mathcal{D}_J^{\mathrm{env\ norm}} := \mathcal{D}_J^{\mathrm{env}} \cdot \frac{\mathcal{D}_J^{\mathrm{med}}}{\mathcal{D}_J^{\mathrm{env\ med}}}$. Finally, we fit all of the normalized data together using the same functional form.

# D   Additional Experimental Results

## D.1   Batch Size Fits $\tilde{B}(\sigma, N)$

Refer to Figure 11.

$$
\begin{aligned}
\texttt{h1-crawl} \quad & \frac{1680.64}{\sigma^{0.30} + 6.01\mathrm{e}7\,\sigma^{0.30}N^{-1.12}} \\[4pt]
\texttt{h1-pole} \quad & \frac{4112.98}{\sigma^{0.24} + 1.45\mathrm{e}1\,\sigma^{0.24}N^{-0.07}} \\[4pt]
\texttt{h1-stand} \quad & \frac{1458.10}{\sigma^{0.27} + 1.33\mathrm{e}74\,\sigma^{0.27}N^{-12.71}} \\[4pt]
\texttt{humanoid-stand} \quad & \frac{1160.40}{\sigma^{0.49} + 2.77\mathrm{e}2\,\sigma^{0.49}N^{-0.38}}
\end{aligned}
\tag{D.1}
$$

## D.2   Batch Size Fit Analysis

In Table 5, we group runs by UTD and model size, and bin runs based on batch sizes. Then, we consider the data efficiency ratio between the runs appearing in bins with suboptimal batch sizes and runs with the predicted batch size, and average over UTDs and model sizes. We find that batch sizes within a interval around the best batch size $B^*$ perform reasonably, and performance degrades significantly with larger intervals. Indeed, per this analysis, *one cannot naïvely reuse the same batch size for small and large models*: in Figure 5, we see a $\approx 40\times$ range in bootstrap-optimal batch sizes across different model sizes at UTD 8. However, the sensitivity of performance to the precise value of batch size is relatively low, which is good news for practitioners and which is why we observe a high relative error in the fit, which turns out to be benign.

## D.3   Learning Rate Sensitivity Analysis

A natural question is whether learning rate affects performance in the compute-optimal regime or not. We found that there is a range of "reasonable" learning rates, which empirically always contains our

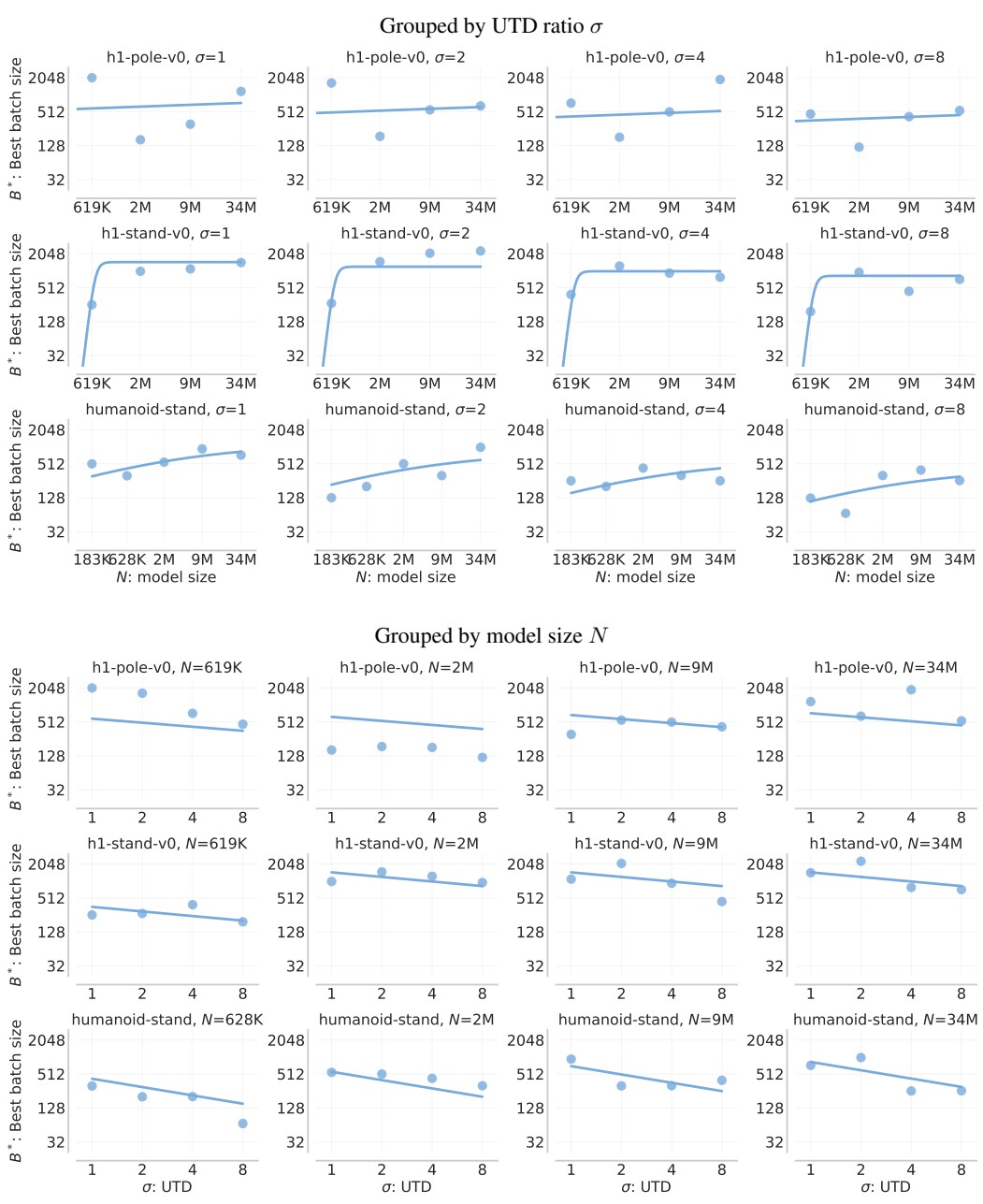

Figure 11: Two-dimensional batch size fit $\tilde{B}(\sigma, N)$ grouped by $\sigma$ and $N$, as a completion to Figure 5, for the BRO algorithm and architecture [34]

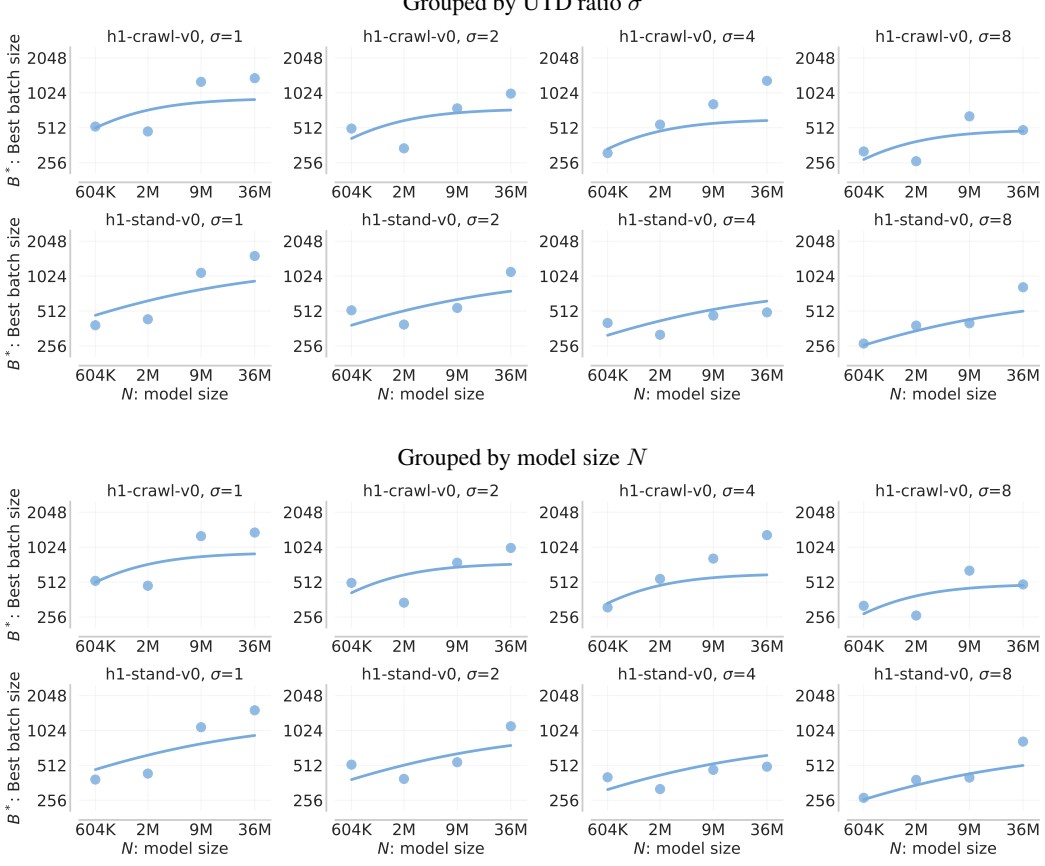

Figure 12: Analogous to Figure 11, with the SimbaV2 architecture [26]

Table 5: Batch size sensitivity over grid search. Batch sizes far away from the predicted batch size perform poorly.

| Batch size range | Data efficiency ratio |
|---|---|
| $[1/16\,B^*, 1/8\,B^*]$ | 1.52 |
| $[1/8\,B^*, 1/4\,B^*]$ | 1.38 |
| $[1/4\,B^*, 1/2\,B^*]$ | 1.26 |
| $[1/2\,B^*, 2/3\,B^*]$ | 1.22 |
| $B^*$ | 1.00 |
| $[1.5\,B^*, 2\,B^*]$ | 1.16 |
| $[2\,B^*, 4\,B^*]$ | 1.18 |
| $[4\,B^*, 8\,B^*]$ | 1.19 |
| $[8\,B^*, 16\,B^*]$ | 1.30 |

Table 6: Learning rate sensitivity over grid search.

| Learning rate range | Data efficiency ratio |
|---|---|
| $[1/4\,\mathrm{lr}^*, 1/2\,\mathrm{lr}^*]$ | 1.39 |
| $[1/2\,\mathrm{lr}^*, 2/3\,\mathrm{lr}^*]$ | 1.35 |
| $[2/3\,\mathrm{lr}^*, \mathrm{lr}^*]$ | 1.04 |
| $[\mathrm{lr}^*, 1.5\,\mathrm{lr}^*]$ | 1.03 |
| $[1.5\,\mathrm{lr}^*, 2\,\mathrm{lr}^*]$ | 1.18 |
| $[2\,\mathrm{lr}^*, 4\,\mathrm{lr}^*]$ | 1.27 |

Table 7: Bootstrap-optimal vs. default learning rates over compute-optimal $(\sigma, N, B)$.

| Environment | Data efficiency ratio |
|---|---|
| h1-crawl | 1.0118 |
| h1-pole | 1.0000 |
| h1-stand | 1.0000 |
| humanoid-stand | 0.9504 |

"default" value of 3e-4. ***Crucially, this is the case for all model sizes and UTD ratio,*** meaning that a practitioner can get away without setting learning rate carefully for a compute-optimal run as long as they utilize a default value.

**Grid search regime.** We run hyperparameter sweeps over (model size, UTD, learning rate) and (model size, batch size, learning rate), where lr $\in \{$1e-4, 2e-4, 3e-4, 6e-4$\}$. In this regime, we found that the empirically optimal learning rate only took on values $\{$2e-4, 3e-4$\}$. We report the data efficiency ratio between the empirically optimal and default learning rates in Table 6. Since our default learning rate is in the range $[\mathrm{lr}^*, 1.5\,\mathrm{lr}^*]$, the overall effect on performance is minimal.

Although we observe smaller relative variation in the best learning rate over UTD and model sizes compared to batch size, we find empirically that the best learning rate (i) decreases with increasing model size, correlation: -0.75, (ii) decreases with increasing UTD, correlation: -0.46, (iii) increases with increasing batch size, correlation: 0.42. With simple log-linear fits, we obtain a relative error of 37.5%:

$$\texttt{h1-crawl} \qquad \mathrm{lr}^* \sim 4.4827\text{e-}4 \cdot (N/2.3\text{e}6)^{-0.3112} \cdot \sigma^{-0.1273} \cdot (B/512)^{0.3709}$$

$$\texttt{h1-pole} \qquad \mathrm{lr}^* \sim 2.4727\text{e-}4 \cdot (N/2.3\text{e}6)^{-0.2472} \cdot \sigma^{-0.2392} \cdot (B/512)^{0.2701}$$

Despite the high relative error, we observe that data efficiency is similar within an interval of "reasonable" learning rates.

**Compute-optimal regime.** For each task and compute-optimal setting $\sigma^*(\mathcal{C}_0), N^*(\mathcal{C}_0)$ with fitted batch size $\tilde{B}(\sigma^*(\mathcal{C}_0), N^*(\mathcal{C}_0))$, we ran a sweep of learning rates over [1e-4, 2e-4, 3e-4, 4e-4, 5e-4]. Following Eq. (C.1), we compute the bootstrap-optimal learning rate for each setting, then round to the nearest of the five learning rates. In Table 7, we show that data efficiency is not improved significantly when using the rounded bootstrap-optimal learning rate, compared to the "default" learning rate 3e-4. The table shows averages over compute budgets.

### D.4 Target Network Update Rate Sensitivity Analysis

Value-based deep RL methods train a Q-network $Q_\theta$ by minimizing the TD-error against the target Q-network $\bar{Q}$ (Eq. (3.1)). The target network weights $\bar{\theta}$ are typically updated via Polyak averaging, $\bar{\theta} \leftarrow (1 - \tau)\bar{\theta} + \tau\theta$, where $\tau$ is a constant, the target network update rate. Small $\tau$ yield high-bias, low-variance targets; large $\tau$ the opposite. Intuitively, $\tau$ seems to be an important hyperparameter for modulating the dynamics of TD-learning. Empirically, however, we do not find a strong relationship between the model size, the target network update rate $\tau$, and training or validation TD error. We ran a sweep over $\tau \in [$5e-4, 1e-3, 2e-3, 5e-3, 1e-2, 2e-2, 5e-2, 1e-1, 2e-1$]$. Then, we fit a power law TD error $\sim a \cdot \tau^b$, and record the correlation and slope in Table 8. In general, we find that training and validation TD error increase with $\tau$ (positive slope and correlation), but there is not a strong relationship between model size and the corresponding correlation or slope.

Table 8: Correlation between target update rate $\tau$

| Task | Metric | Critic width | Correlation | Slope ($b$) |
|------|--------|-------------|-------------|-------------|
| h1-crawl | Critic loss | 512 | 0.7365 | 0.1203 |
| | | 1024 | 0.9175 | 0.1348 |
| | | 2048 | 0.9302 | 0.1164 |
| | Validation critic loss | 512 | 0.4639 | 0.0329 |
| | | 1024 | 0.6413 | 0.0244 |
| | | 2048 | 0.4751 | 0.0168 |
| h1-stand | Critic loss | 512 | 0.9056 | 0.3916 |
| | | 1024 | 0.9446 | 0.2035 |
| | | 2048 | 0.1857 | 0.0195 |
| | Validation critic loss | 512 | 0.7777 | 0.1294 |
| | | 1024 | 0.3109 | 0.0196 |
| | | 2048 | -0.6852 | -0.0421 |

We additionally found that data efficiency is not very sensitive to our choice of $\tau$, as long as the value of $\tau$ is reasonable. Following the same sensitivity analysis from Section 6, we find that varying $\tau$ by *an order of magnitude* from the bootstrapped optimal value of $\tau$ worsens the data efficiency by only 19%. For comparison, varying the batch size by an order of magnitude yields a data efficiency variation of up to 52% (Table 5). Throughout the remainder of our experiments, we use a "default" value of $\tau = $ 5e-3, which we find is within the "reasonable" interval and near the bootstrapped optimal value.

### D.5   Full TD-error curves

We provide full training and validation TD-error curves in Figure 13, as a completion to Figure 1. The summary statistics are marked with 'X' and correspond to the points used in Figure 2.

### D.6   Passive Critic Learning Curves

We provide the full validation TD error curves over training in Figure 14. In these plots the summary statistics are marked with 'X', and we provide Figure 15 as a completion to Figure 4.

### D.7   Data Efficiency Fits $\mathcal{D}_{J_{\max}}(\sigma, N)$

For the following four tasks, we fit data efficiency using the empirically best data efficiency for performance threshold $J_{\max}$ across batch sizes for each $(\sigma, N)$ setting. In Figures 16 and 17, we show the fits for multiple values of $J$.

$$
\begin{aligned}
\texttt{h1-crawl} \quad & 5.11\mathrm{e}4 \left(1 + (2.59\mathrm{e}5/\sigma)^{0.15} + (1.70\mathrm{e}7/N)^{0.75}\right) \\
\texttt{h1-pole} \quad & 9.43\mathrm{e}3 \left(1 + (3.22\mathrm{e}6/\sigma)^{0.27} + (2.50\mathrm{e}12/N)^{0.30}\right) \\
\texttt{h1-stand} \quad & 2.14\mathrm{e}5 \left(1 + (6.68\mathrm{e}\text{-}1/\sigma)^{2.53} + (1.74\mathrm{e}6/N)^{0.97}\right) \\
\texttt{humanoid-stand} \quad & 1.49\mathrm{e}4 \left(1 + (1.78\mathrm{e}6/\sigma)^{0.26} + (3.75\mathrm{e}7/N)^{0.63}\right)
\end{aligned}
\tag{D.2}
$$

For the remaining tasks, we use the available batch size.

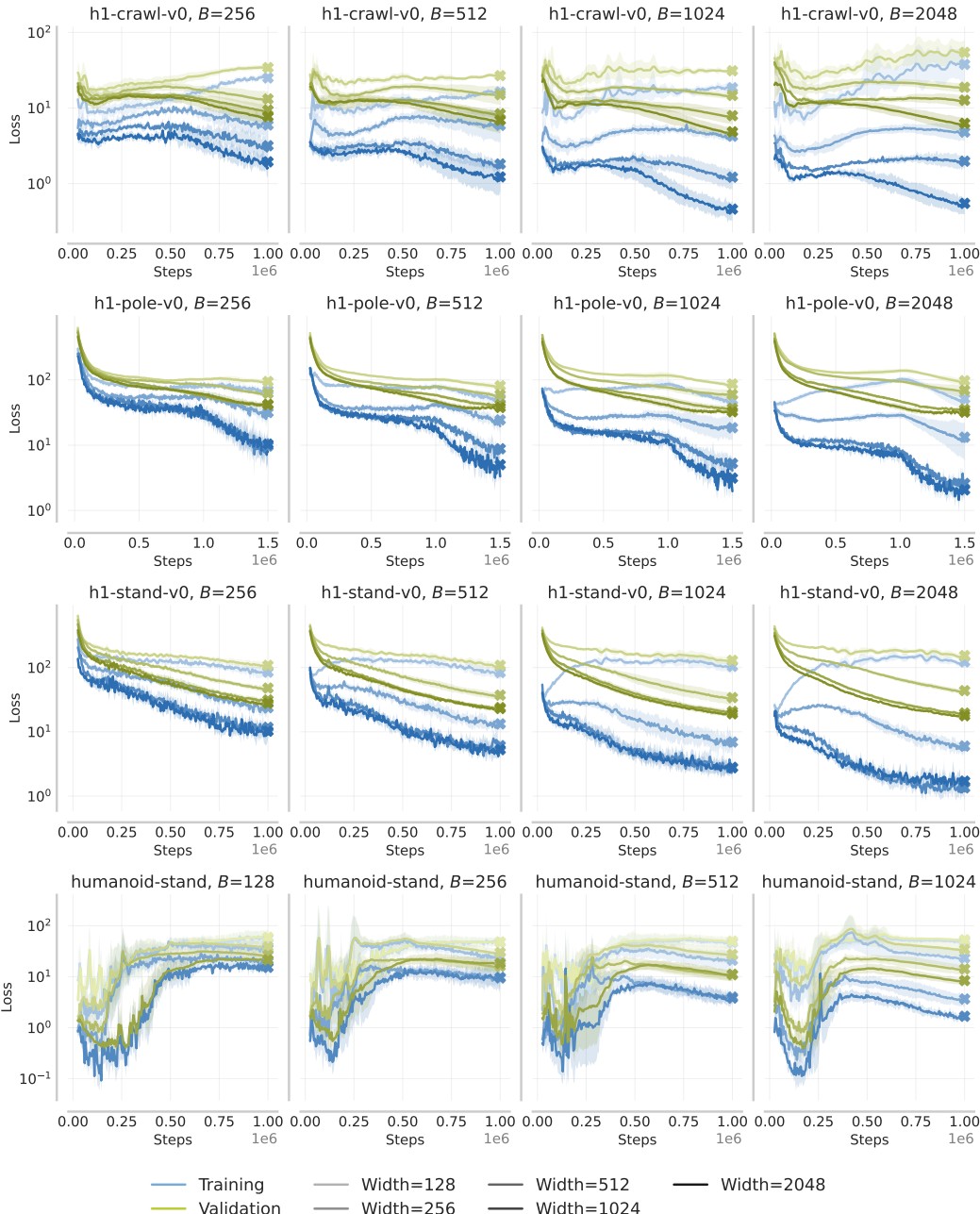

Figure 13: Training and validation TD-error curves over training, grouped by critic width and passive critic width, at UTD = 1. The summary statistics in Figure 2 are marked with 'X' and are averages over the last 10% of training.

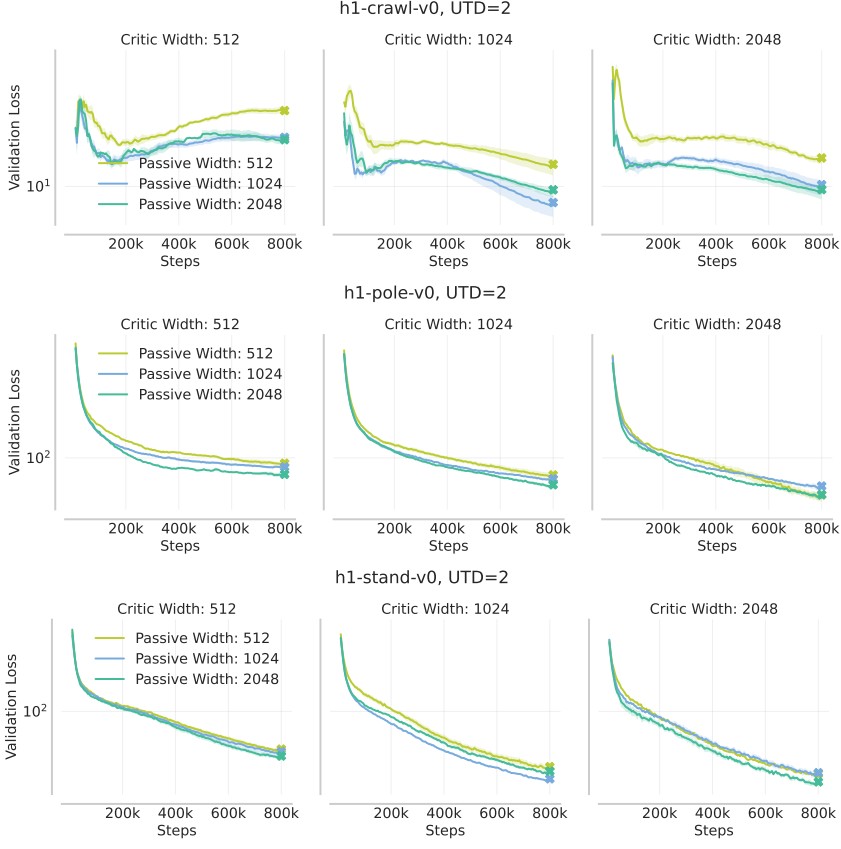

Figure 14: Validation TD-error curves over training, grouped by critic width. The summary statistics in Figure 4 are marked with 'X' and are averages over the last 10% of training.

**DMC-medium, shared $\alpha_J$, $\beta_J$:**

$$
\begin{aligned}
\texttt{acrobot-swingup} \quad & 4.29\text{e}5\left(1 + (6.46\text{e-}1/\sigma)^{0.98} + (8.42\text{e}5/N)^{1.39}\right) \\
\texttt{cheetah-run} \quad & 4.81\text{e}5\left(1 + (4.30\text{e-}1/\sigma)^{0.98} + (3.40\text{e}5/N)^{1.39}\right) \\
\texttt{finger-turn} \quad & 2.66\text{e}5\left(1 + (1.08\text{e}0/\sigma)^{0.98} + (4.67\text{e}5/N)^{1.39}\right) \\
\texttt{fish-swim} \quad & 6.28\text{e}5\left(1 + (1.36\text{e-}1/\sigma)^{0.98} + (2.70\text{e}5/N)^{1.39}\right) \qquad \text{(D.3)} \\
\texttt{hopper-hop} \quad & 3.52\text{e}5\left(1 + (8.24\text{e-}1/\sigma)^{0.98} + (3.12\text{e}5/N)^{1.39}\right) \\
\texttt{quadruped-run} \quad & 1.39\text{e}5\left(1 + (3.54\text{e}0/\sigma)^{0.98} + (1.64\text{e}6/N)^{1.39}\right) \\
\texttt{walker-run} \quad & 1.61\text{e}5\left(1 + (2.85\text{e}0/\sigma)^{0.98} + (6.07\text{e}5/N)^{1.39}\right)
\end{aligned}
$$

**DMC-medium, averaged environment:**

$$
\texttt{DMC-medium averaged} \quad 3.72\text{e}5\left(1 + (1.26\text{e}0/\sigma)^{1.01} + (6.33\text{e}5/N)^{0.89}\right) \qquad \text{(D.4)}
$$

**DMC-hard, shared $\alpha_J$, $\beta_J$.**

$$
\begin{aligned}
\texttt{dog-run} \quad & 4.45\text{e}5\left(1 + (1.23\text{e}0/\sigma)^{0.73} + (9.26\text{e}5/N)^{1.29}\right) \\
\texttt{dog-stand} \quad & 4.40\text{e}5\left(1 + (1.94\text{e-}1/\sigma)^{0.73} + (3.94\text{e}5/N)^{1.29}\right) \\
\texttt{dog-trot} \quad & 5.38\text{e}5\left(1 + (6.42\text{e-}1/\sigma)^{0.73} + (7.53\text{e}5/N)^{1.29}\right) \\
\texttt{dog-walk} \quad & 6.03\text{e}5\left(1 + (3.09\text{e-}1/\sigma)^{0.73} + (3.78\text{e}5/N)^{1.29}\right) \qquad \text{(D.5)} \\
\texttt{humanoid-run} \quad & 4.29\text{e}5\left(1 + (2.04\text{e}0/\sigma)^{0.73} + (1.00\text{e}6/N)^{1.29}\right) \\
\texttt{humanoid-walk} \quad & 3.30\text{e}5\left(1 + (3.81\text{e}0/\sigma)^{0.73} + (1.13\text{e}6/N)^{1.29}\right)
\end{aligned}
$$

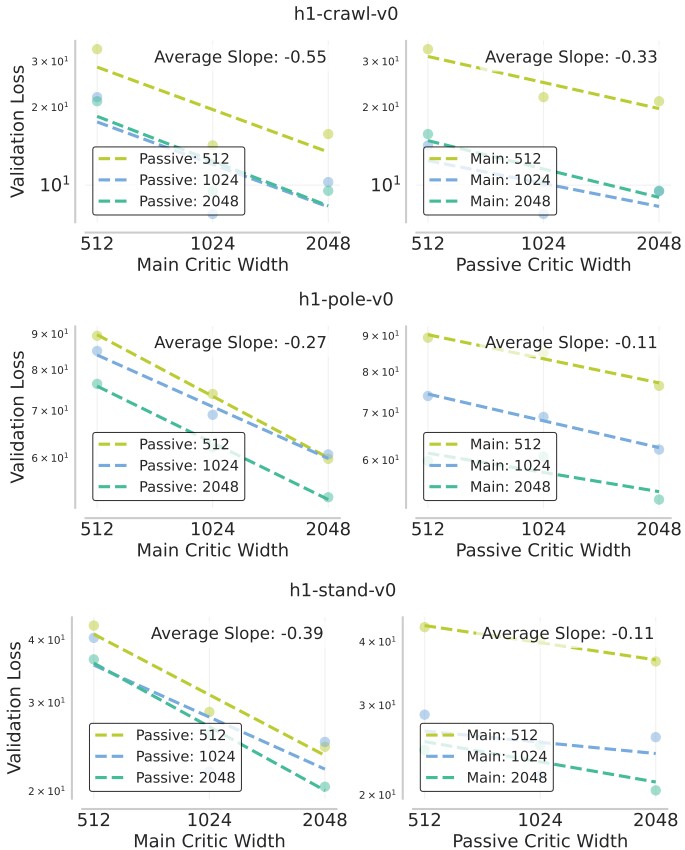

Figure 15: Summary statistics for passive critic experiments, as a completion of Figure 4, run at UTD 2. Across multiple environments, increasing the main critic size is much more effective than increasing the passive critic size.

**DMC-hard, averaged environment:**

$$\texttt{DMC-hard averaged} \qquad 5.39\text{e}5 \left(1 + (8.92\text{e-}1/\sigma)^{0.77} + (6.68\text{e}5/N)^{1.27}\right) \qquad \text{(D.6)}$$

## D.8 Optimal Budget Partition

We provide plots analogous to Figures 7 and 8 in Figures 18 to 21 for DMC-medium and DMC-hard tasks. These data efficiency fits use the shared exponents $\alpha_J$, $\beta_J$ method described in Appendix B.

As shown in Figures 8, 20 and 21, however, the optimal UTD and model size for a given budget $\mathcal{F}_0$ are unpredictable. We verify that these hyperparameters are fundamentally unpredictable in this setting, running at least 50 seeds per UTD and model size at the fitted batch size, for h1-crawl, in Figure 22. Despite this, the fit for $\mathcal{C}_{\mathcal{F}_J}$ achieves considerably lower uncertainty than in Figure 7, indicating that there is a large range of "reasonable" hyperparameters corresponding to similar data and compute values.

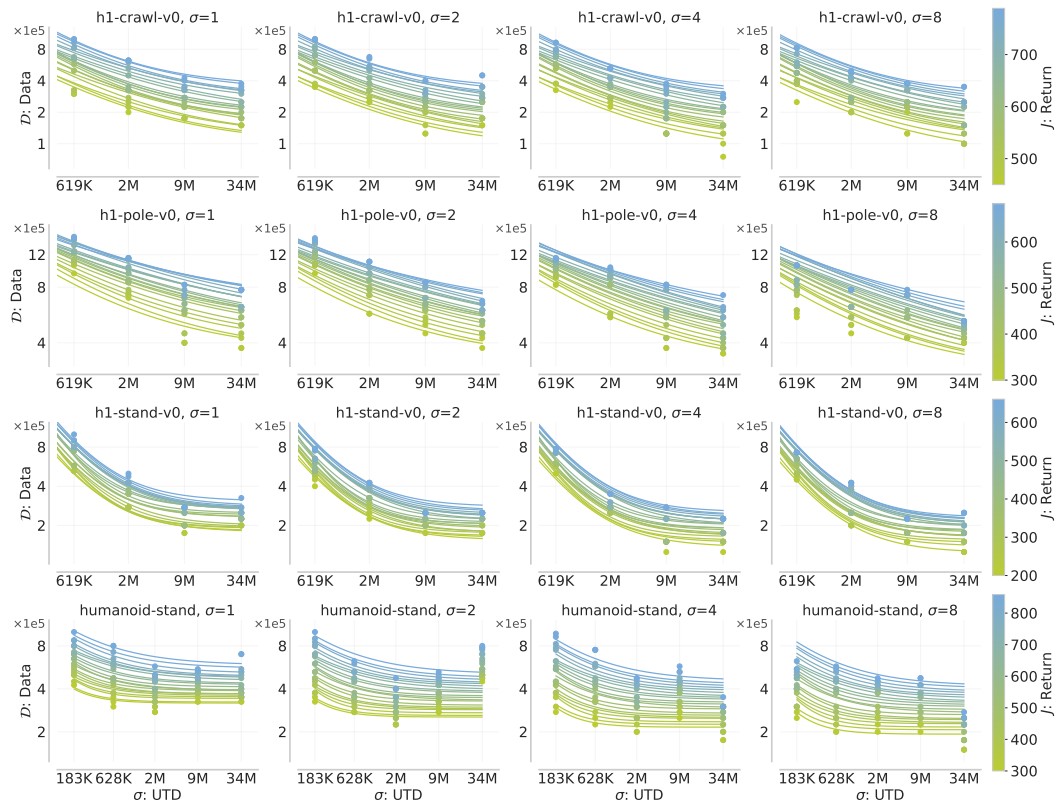

Figure 16: Data efficiency fits $\mathcal{D}_J(\sigma, N)$ for multiple performance thresholds $J$, grouped by UTD ratio $\sigma$. Each $\mathcal{D}_J$ is fit independently.

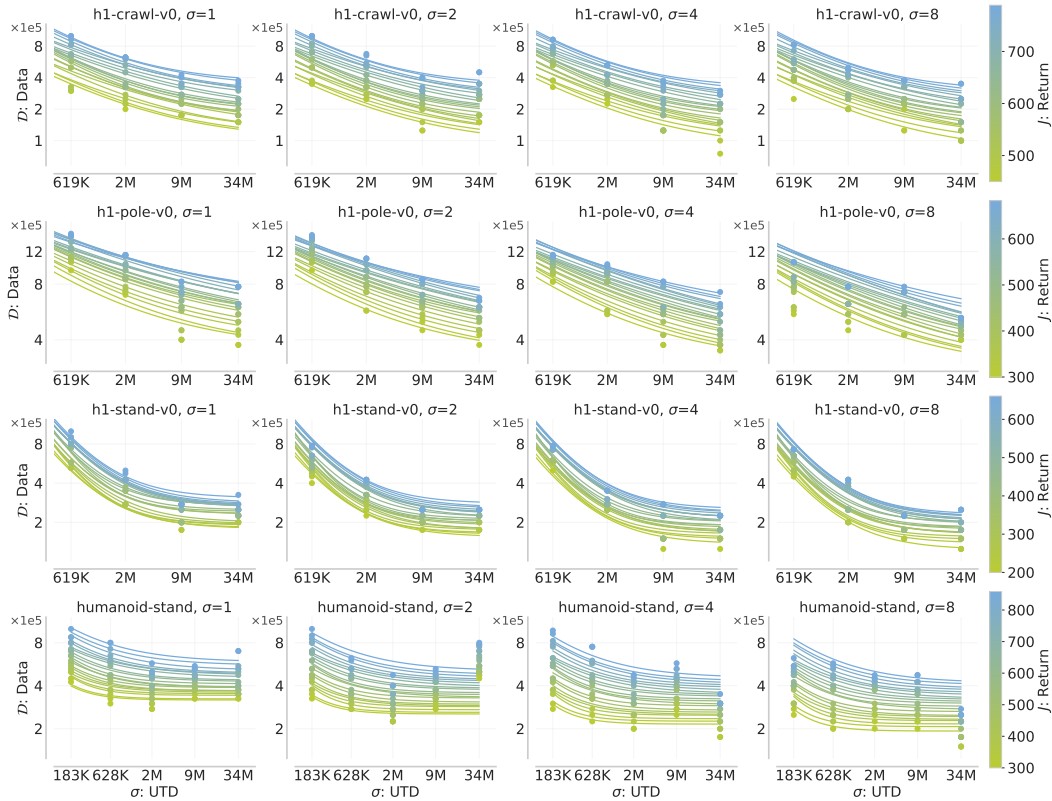

Figure 17: Same as Figure 16, but instead grouped by model size $N$.

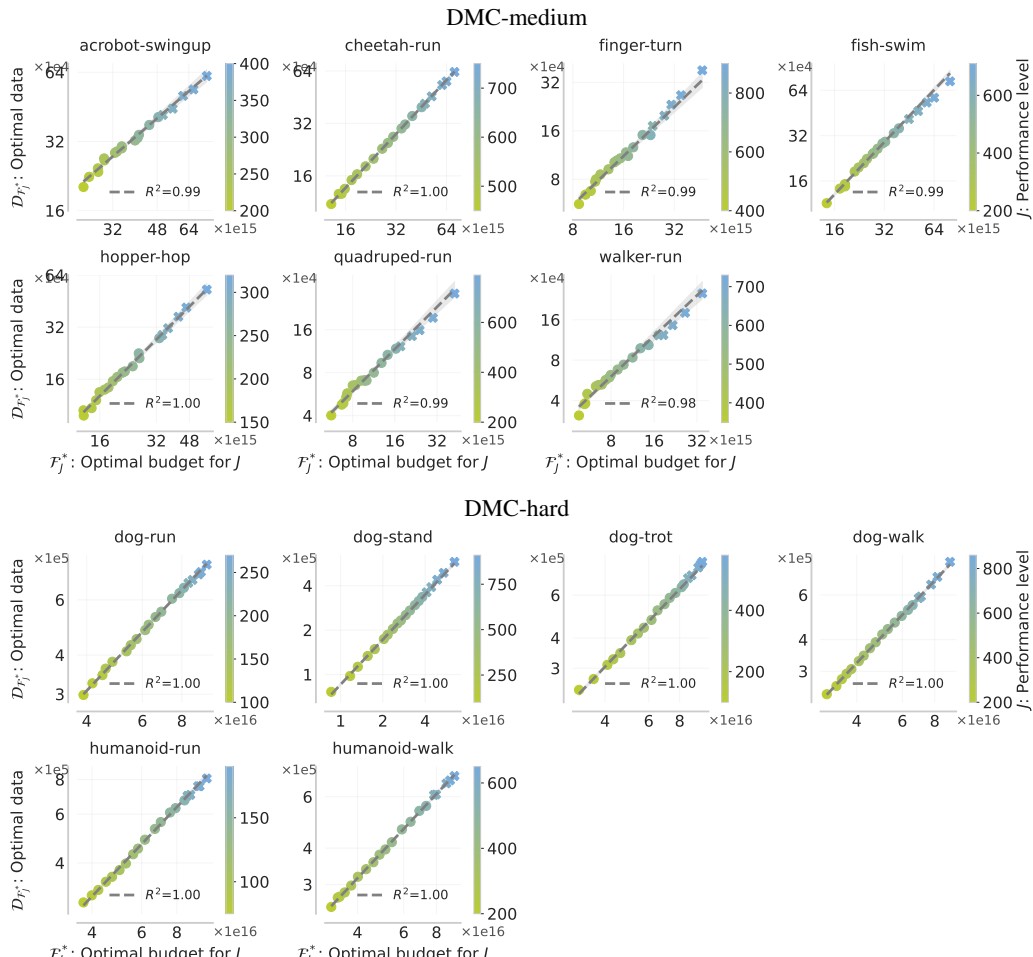

Figure 18: Optimal data $\mathcal{D}(\mathcal{F}_0)$ for a given budget $\mathcal{F}_0$, as a completion of Figure 7.

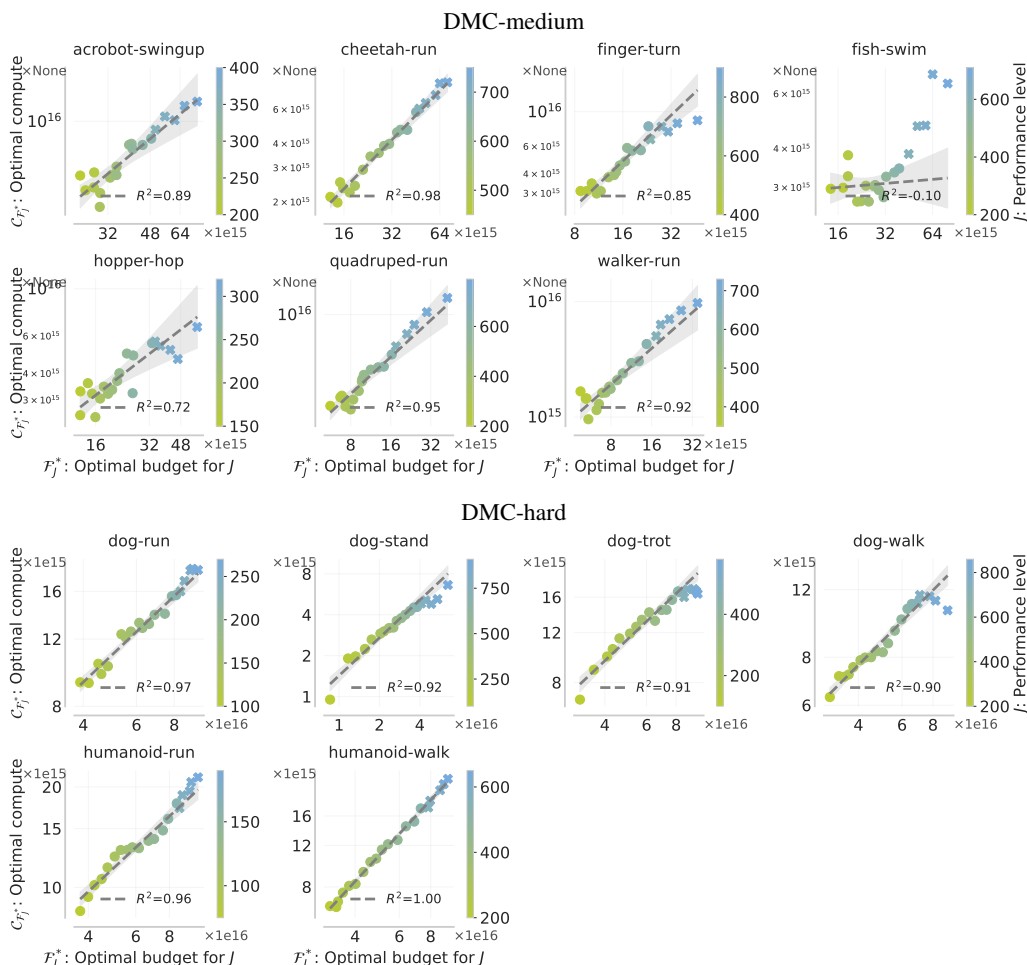

Figure 19: Optimal compute $\mathcal{C}(\mathcal{F}_0)$ for a given budget $\mathcal{F}_0$, as a completion of Figure 7.

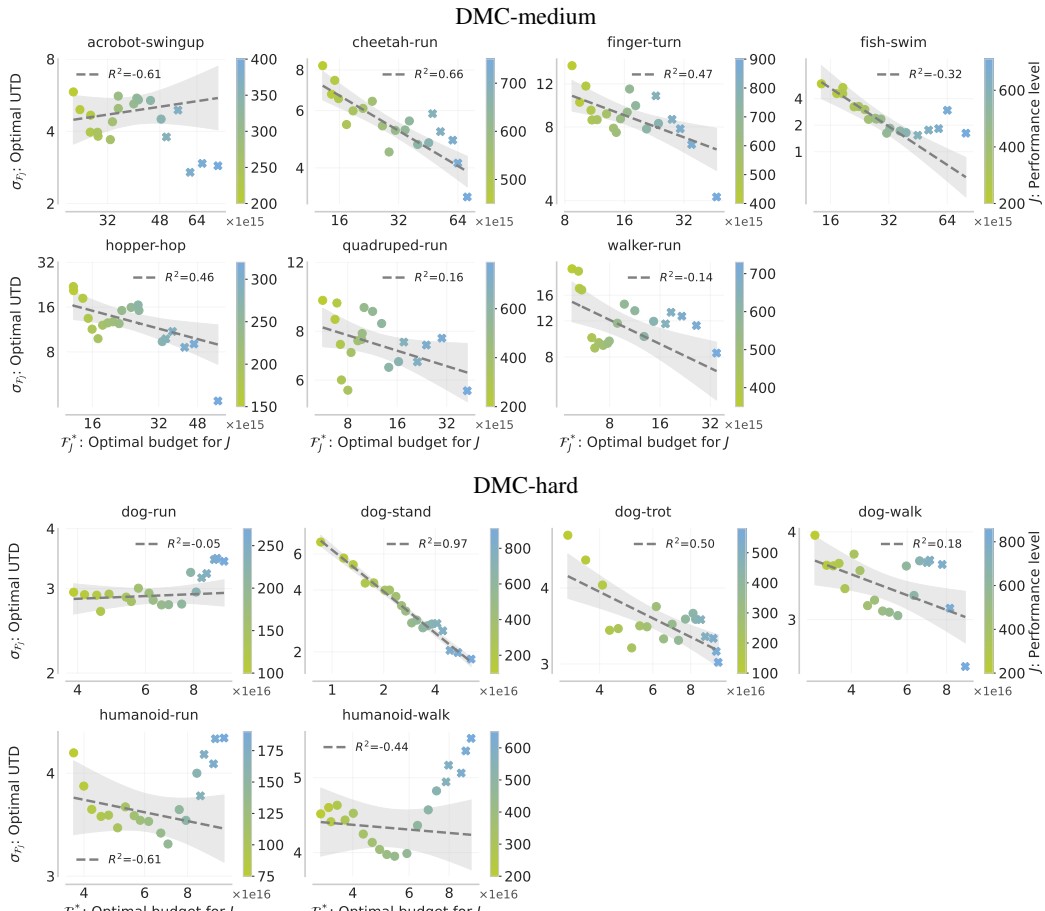

Figure 20: Optimal UTD ratio $\sigma_{\mathcal{F}}^*(\mathcal{F}_0)$ for a given budget $\mathcal{F}_0$, as a completion of Figure 8.

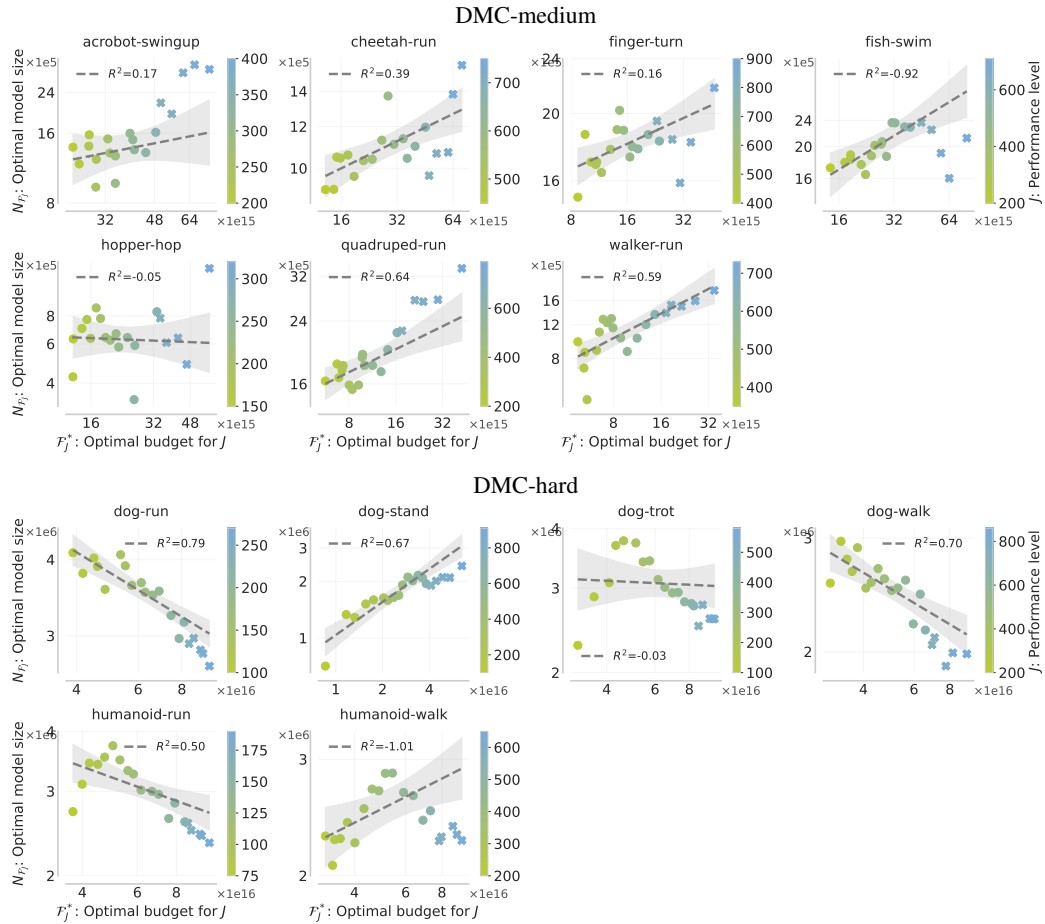

Figure 21: Optimal model size $N_{\mathcal{F}}^*(\mathcal{F}_0)$ for a given budget $\mathcal{F}_0$, as a completion of Figure 8.

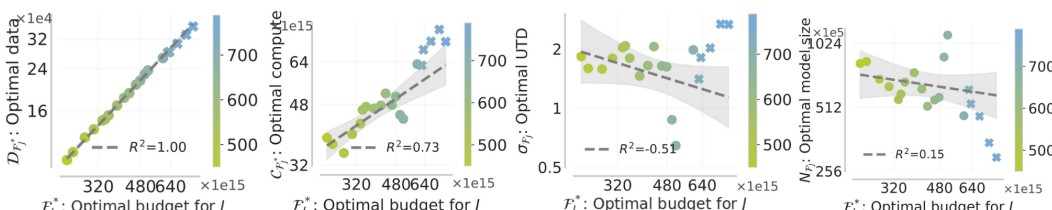

Figure 22: Optimal data, compute, UTD, and model size for a given budget $\mathcal{F}_0$, run for 50+ seeds on `h1-crawl`.

