# OpenReview forum: "Compute-Optimal Scaling for Value-Based Deep RL"
_NeurIPS.cc/2025/Conference — NeurIPS 2025 poster_

### Official Review · Reviewer_R4Hr · 2025-06-17

**Clarity:** 2
**Significance:** 3
**Originality:** 3
**Rating:** 5
**Confidence:** 3

**Summary:**

This paper presents a systematic study on how model size, UTD, and batch size affect the performance of value-based deep reinforcement learning (RL) algorithms.
The key insight is the identification of "TD-overfitting," where large batch sizes degrade generalization in small models but improve it in larger ones (e.g., Takeaway 2).
Based on empirical observations, the authors propose practical scaling rules for optimal compute allocation across model size, UTD, and batch size (Scaling Observations 1-4).

**Questions:**

See weaknesses.

**Ethical Concerns:**

["NO or VERY MINOR ethics concerns only"]

**Final Justification:**

The response addressed my concerns about the experimental setup (e.g., model architecture, robotic task, and n-step TD).

**Limitations:**

The empirical results are limited to robotic tasks, as described in Section 8. I think this limitation does not diminish the quality of this paper, as long as there is a solid motivation to study robotic tasks.

**Paper Formatting Concerns:**

I have no formatting concern

**Quality:**

3

**Strengths And Weaknesses:**

**Strengths:**

* The paper is well-written, and the empirical results are generally strong.
* The identification of "TD-overfitting" is a novel and insightful contribution. The resulting scaling rules are well-motivated by this finding and appear practical.

**Weaknesses**

The experimental setup lacks clarity and persuasiveness.

=====

Why do the authors focus specifically on the BRO architecture? While I understand that regularization is crucial in practice, it can also influence the behavior of TD error. I am concerned that this may affect the interpretation of the TD-overfitting phenomenon. If regularization significantly alters the behavior, this should be clearly discussed and reflected in the main takeaways.

=====

Why do the authors focus primarily on robotic tasks? To convincingly demonstrate the TD-overfitting phenomenon, it would be more natural to include experiments based on value-iteration-style algorithms, such as DQN. Continuous action tasks should introduce additional approximation errors when evaluating the TD errors compared to discrete action settings, specifically when taking the expectation over $\mathbb{E}_{a \sim \pi}$.

Is there a specific motivation for studying robotic tasks in the context of TD-overfitting? If so, this rationale should be clearly articulated.

=====

The TD-overfitting phenomenon seems to be evaluated only in the 1-step TD setting. Since $\infty$-step TD corresponds to the unbiased return estimator, I expect that multi-step TD may be less susceptible to TD-overfitting than the 1-step version. I think evaluating multi-step TD and analyzing its behavior would offer stronger support for the TD-overfitting phenomenon.

Could the authors provide either theoretical implications or empirical evidence regarding this point?
I believe a small-scale experiment, such as one using MinAtar, would be sufficient to validate the scaling rules in alternative settings.

* Rainbow: Combining Improvements in Deep Reinforcement Learning : https://arxiv.org/abs/1710.02298

---

> ### Author Rebuttal · Authors · 2025-07-31
>
> Thank you for your positive and constructive feedback. To address your concerns, we are including several new experiments and discussion as requested, which we believe further strengthen our submission. In particular, we are adding new results on different architectures beyond BRO, n-step TD, and evaluations of scaling on Atari games, a domain quite distinct from robotics. **Please let us know if these responses address your concerns, and if so, we would be grateful if you are willing to raise your score.**
>
> > BRO architecture?
>
> Our decision to focus on the BRO architecture was motivated by findings in prior work [1,2], which show that naively scaling critic models often leads to degraded performance. BRO incorporates architectural and regularization components (i.e. BroNet) that have been shown to enable stable critic scaling with consistent performance gains. We agree that regularization can influence the behavior of TD error, and chose BRO to study TD overfitting and scaling in a setting where scaling is feasible. Our findings and the fitted scaling laws should therefore be interpreted as conditional on an architecture that supports scalable TD learning. This type of conditional framing is consistent with prior work in model scaling, such as scaling laws for dense transformers [3], mixture-of-experts models [4], or state-space architectures [5], where scaling behavior is reported for architectures known to scale well. To clarify this for readers, we now emphasize in the main text that our scaling laws are specific to the BRO architecture, and the parameters of our fitted curves depend on regularization and architectural design.
>
> > Robotic tasks?
>
> We chose to focus primarily on robotic tasks because our goal is to investigate whether model scaling in value-based RL can yield predictable improvements, similar to trends observed in supervised learning. This is a non-trivial question as prior work [1,2,6,7] has highlighted that naively increasing model size or UTD often leads to unstable or degraded performance in RL. The considered robotic tasks are commonly used in recent RL model scaling studies [2,6], enabling direct comparison and contextualization of our results.
>
> That said, we agree that evaluating scaling behavior in other domains, such as discrete action environments with value-iteration-style algorithms, is an important direction. To this end, we are currently running a subset of Atari-100K experiments with DQN, and preliminary results shown in the table below confirm that model scaling can yield monotonic improvements in performance in discrete-action tasks as well and we will expand this with the TD overfitting analysis in the camera-ready version. Furthermore, we have expanded the limitations section to explicitly state that our current findings may not generalize across all RL settings (e.g. goal-conditioned, sparse reward, or pixel-based tasks), and require further validation.
>
> | Model size |   $J$ = 0.16 |   $J$ = 0.27 |   $J$ = 0.38 | $J$ = 0.49   | $J$ = 0.60   |
> |-----------:|-------------:|-------------:|-------------:|:-------------|:-------------|
> |      150K  |        11880 |        19800 |        38010 | N/A          | N/A          |
> |      320K  |        10880 |        18130 |        26460 | 54130        | 82520        |
> |      790K  |        10310 |        17180 |        24050 | 40580        | 60010        |
> |     2.13M  |         8710 |        14520 |        20330 | 26800        | 36040        |
> |     6.47M  |         7860 |        13100 |        18340 | 23580        | 33270        |
>
> > Continuous action space?
>
> We thank the reviewer for this suggestion. We agree that continuous action tasks introduce additional approximation challenges, particularly when estimating the TD target via action sampling or optimization (as opposed to exact maximization in discrete settings). This does make theoretical analysis of TD error more complex. However, our main motivation for focusing on robotic control tasks is to study TD overfitting in high-dimensional, realistic settings where RL is frequently deployed, e.g. in applications such as sim-to-real. To this end, we include experiments on challenging, high-dimensional environments, such as the simulated Unitree robot from HumanoidBench, to ground our analysis in realistic control settings. We agree complementing this with discrete action experiments would offer a useful theoretical contrast, and we are working toward that in parallel (see our response above on Atari-100K).
>
> > n-step TD?
>
> We thank the reviewer for this insightful suggestion. We agree that, theoretically, increasing $n$ in multi-step TD should reduce the susceptibility to TD-overfitting - as $n \rightarrow \infty$, the learning target approaches a Monte-Carlo return estimate, effectively turning the problem into supervised regression on full returns rather than bootstrapped targets. In this sense, 1-step TD represents the most bootstrapped and thus most overfitting-prone setting. However, in the off-policy setting, multi-step TD introduces additional bias, especially when learning from older transitions that reflect outdated or suboptimal policies. In practice, this tradeoff between bias and variance can confound the expected reduction in TD-overfitting as well. To this end, our preliminary experiments with the BRO architecture suggested that n-step TD improves performance with small models, where variance reduction helps stability. However, performance at large n degrades with larger models, likely due to increased off-policy bias. We are currently running more comprehensive experiments to validate these observations and plan to include these results in the final version.
>
>
> [1] Bjorck, Nils, et al. "Towards deeper deep reinforcement learning with spectral normalization." NeurIPS 2021
>
> [2] Nauman, Michal, et al "Bigger, Regularized, Optimistic." NeurIPS 2024
>
> [3] Kaplan, Jared, et al. “Scaling Laws for Neural Language Models”. arXiv 2020
>
> [4] Krajewski, Jakub, et al. "Scaling laws for fine-grained mixture of experts." arXiv 2024
>
> [5] Gu, Albert and Dao Tri. "Mamba: Linear-Time Sequence Modeling with Selective State Spaces." COLM 2024
>
> [6] Lee, Hojoon, et al. "Simba: Simplicity bias for scaling up parameters in deep reinforcement learning." ICLR 2025
>
> [7] D'Oro, Pierluca, et al. "Sample-Efficient Reinforcement Learning by Breaking the Replay Ratio Barrier." ICLR 2023

---

### Official Review · Reviewer_Vd6A · 2025-06-27

**Clarity:** 3
**Significance:** 4
**Originality:** 4
**Rating:** 5
**Confidence:** 5

**Summary:**

This paper investigates optimal compute allocation strategies for value-based deep reinforcement learning methods using temporal-difference (TD) learning. The authors examine two primary scaling dimensions: model size and the updates-to-data (UTD) ratio. Through controlled experiments, they identify a novel phenomenon called "TD-overfitting" where smaller models exhibit poor generalization (higher validation TD-error) when trained with large batch sizes, while larger models benefit from increased batch sizes. The authors trace this phenomenon to poor-quality TD-targets generated by smaller networks and validate this hypothesis through passive critic experiments. Based on these insights, they develop empirical models for optimal batch size selection and provide guidance for jointly scaling model size and UTD ratio given fixed compute budgets.

**Questions:**

- Learning rate scaling: In supervised learning, learning rates are typically scaled proportionally with batch size (e.g., linear scaling rule). However, this study appears to use fixed learning rates across different batch sizes. Would the optimal batch size curves improve if learning rates were scaled proportionally with batch size? How might this affect the TD-overfitting threshold?
- Double descent phenomenon: Supervised learning typically exhibits double descent, where performance initially degrades with increased model size before improving again. However, in your experiments, performance appears to improve gradually with model size without clear degradation phases. Have you observed any double descent phenomena in your scaling experiments across different batch sizes or UTD ratios? If not, what might explain the absence of this commonly observed pattern in the TD-learning setting?

**Ethical Concerns:**

["NO or VERY MINOR ethics concerns only"]

**Final Justification:**

This paper provides a compelling TD-overfitting hypothesis that reconciles batch size practices across RL domains and offers practical scaling guidance. While its generality beyond BRO is uncertain, it makes a strong contribution to NeurIPS.

**Limitations:**

- Offline RL applicability: The current analysis focuses on online RL settings where data collection and training occur simultaneously. In offline RL setups where the data budget is fixed and constrained, would these scaling relationships hold identically? The dynamics of TD-overfitting might differ when the dataset is static and finite.
- Performance-optimal vs. compute-optimal scaling: In online RL with simulators, practitioners often prioritize finding the best performance while treating data collection as relatively inexpensive (assuming unlimited environment interactions). In such scenarios, how would one determine the optimal batch size and UTD ratio that maximizes final performance rather than compute efficiency? Would the identified scaling laws apply when the objective shifts from compute-optimal to performance-optimal training?
- Extension to self-supervised learning: The TD-overfitting phenomenon stems from training on self-generated moving targets, which is not unique to RL. Self-supervised learning methods like BYOL, MoCO, and other contrastive approaches also rely on evolving target representations. The paper doesn't investigate whether similar "self-target-overfitting" occurs in these domains, limiting our understanding of how broadly this phenomenon applies to learning systems with non-stationary self-generated targets.

**Paper Formatting Concerns:**

none. Nice figures :)

**Quality:**

4

**Strengths And Weaknesses:**

Strengths:
- Significant theoretical contribution: The TD-overfitting hypothesis provides a compelling explanation for seemingly contradictory practices in the RL community. While Atari-based studies typically employ very small batch sizes (e.g., 32), recent work on continuous control benchmarks using modern architectures like BRO, Simba, and MR.Q successfully leverages much larger batch sizes. The authors' insight that model capacity determines the optimal batch size regime elegantly reconciles these apparently conflicting approaches and suggests that architectural advances have enabled more aggressive batch size scaling.
- Practical impact: This work directly addresses a critical challenge for practitioners who need to efficiently allocate compute resources when scaling RL training. The empirical models for joint optimization of model size and UTD ratio provide actionable guidance for real-world applications.

Weaknesses:
- Limited architectural generalization: While the focus on BRO provides depth, the transferability to other value-based architectures remains unclear. Different network designs, normalization schemes, and update mechanisms could alter the observed scaling relationships.
- Broader implications for self-supervised learning: If the core issue stems from fitting to self-generated moving targets, this phenomenon might extend beyond RL to self-supervised learning methods like BYOL or MoCo, which also train on targets generated by their own evolving representations. The authors don't explore whether similar "self-target-overfitting" occurs in these domains, which could strengthen the theoretical foundation and broaden the impact of their insights.

---

> ### Author Rebuttal · Authors · 2025-07-31
>
> Thank you for the feedback and a positive assessment of our work. To address your concerns, we performed new experiments on alternative network designs for the critic and studied scaling of learning rates for training as reported below. Further, we would like to thank you for additional comments on self-supervised learning and double descent, which look at our work from new and interesting perspectives – we will make sure to add these into our discussion section. In this paper, we will focus on strengthening the existing claims, however, these alternative perspectives might lead to wider impact in future work.
>
> > Different network designs
>
> We ran 5 Atari-100K tasks with DQN with a 2-layer MLP at UTD=1, and for each $J$ estimated the data efficiency for the best batch size. Our preliminary results show that data efficiency improves with model size on average.
>
> | Model size |   $J$ = 0.16 |   $J$ = 0.27 |   $J$ = 0.38 | $J$ = 0.49   | $J$ = 0.60   |
> |-----------:|-------------:|-------------:|-------------:|:-------------|:-------------|
> |      150K  |        11880 |        19800 |        38010 | N/A          | N/A          |
> |      320K  |        10880 |        18130 |        26460 | 54130        | 82520        |
> |      790K  |        10310 |        17180 |        24050 | 40580        | 60010        |
> |     2.13M  |         8710 |        14520 |        20330 | 26800        | 36040        |
> |     6.47M  |         7860 |        13100 |        18340 | 23580        | 33270        |
>
> We agree that regularization can influence the behavior of TD error, and chose BRO to study TD overfitting and scaling in a setting that has shown consistent performance gain with critic scaling. Our findings and the fitted scaling laws should therefore be interpreted as conditional on an architecture that supports scalable TD learning. This type of conditional framing is consistent with prior work in model scaling, such as scaling laws for dense transformers [1], mixture-of-experts models [2], or state-space architectures [3], where scaling behavior is reported for architectures known to scale well. To clarify this for readers, we now emphasize in the main text that our scaling laws are specific to the BRO architecture, and the parameters of our fitted curves depend on regularization and architectural design.
>
> > Learning rate scaling
>
> We have now run an additional analysis of learning rate dependencies. We find that (i) best learning rate decreases with increasing model size, correlation: -0.75, (ii) best learning rate decreases with increasing UTD, correlation: -0.46, (iii) best learning rate increases with increasing batch size, correlation: 0.42. These results are consistent with existing literature McCandlish’18, Yang’22, Rybkin’25. We are running additional experiments to evaluate this hypothesis and will report back before the end of the discussion period.
>
> Below, we show simple log-linear fits of the best learning rate. We empirically find that the best learning scales less than linearly with the batch size.
>
> h1-crawl-v0:
> $R^2$ = 0.3417
> Relative error: 43.5702%
> $\text{lr}^* ~ 4.4827\text{e}-4 \cdot (N/2.3\text{e}6)^{-0.3112} \cdot \sigma^{-0.1273} \cdot (B/512)^{0.3709}$
>
> h1-stand-v0:
> $R^2$ = 0.5702
> Relative error: 32.2800%
> $\text{lr}^* ~ 2.4727\text{e}-4 \cdot (N/2.3\text{e}6)^{-0.2472} \cdot \sigma^{-0.2392} \cdot (B/512)^{0.2701}$
>
> > Performance-optimal vs. compute-optimal scaling
>
> Our framework addresses this issue of finding the “reasonable batch size and optimal UTD ratio to attain the best final performance” as described in Problem 3.1, part 2 and Appendix A.3. Specifically, our framework can be applied in both ways: (i) given a certain performance, minimize the budget required to reach it, and (ii) given a certain budget, maximize performance that can be achieved. Both of these applications can be derived from the functional form of the scaling law that we study. Practitioners can solve for either question as they find necessary, we do not intend to imply that one question is more important than the other.
>
> > self-supervised learning methods like BYOL or MoCo
>
> In this paper, we focus on a studying of SOTA value-based RL methods, which we believe is crucial for future work on large-scale RL. It is an interesting point that self-supervised learning can have some similarities with the analysis proposed in our paper. We would like to note that in value-based RL, an additional source of potential instability arises from the fact the actor is updated continuously, so the values change even faster and due to more factors than in self-supervised learning. It is likely that the value-based RL setting is more challenging. However, self-supervised learning is more mature and allows for larger-scale experiments than value-based RL, and so is an exciting direction of future work.
>
> > Double descent phenomenon
>
> This is a great question! We have not observed a double descent phenomenon in the range of model sizes we evaluated. We would like to note that two complications appear in value-based RL that might impact the double descent presence. First, the notion of overfitting is different in value-based RL since overfitting can occur due to noise in the targets, as discussed in our paper. Second, the notion of network capacity is different, since the network must not only fit the targets at the end of training, but also all targets as they change during training, requiring more capacity. Because of this, further work is needed to analyse whether double descent is present in value-based RL.
>
> > Offline RL applicability
>
> We expect that some of our findings transfer directly to offline RL as the phenomenon of TD-overfitting should be similarly present. However, due to increased distribution shift between the data and the policy, offline RL has other complicating factors such as the need to tune a ‘conservatism’ coefficient or a BC regularization loss. Addressing this is an exciting direction of future work. We will add this to the discussion section of the paper.
>
> [1] Kaplan, Jared, et al. “Scaling Laws for Neural Language Models”. arXiv 2020
>
> [2] Krajewski, Jakub, et al. "Scaling laws for fine-grained mixture of experts." arXiv 2024
>
> [3] Gu, Albert and Dao Tri. "Mamba: Linear-Time Sequence Modeling with Selective State Spaces." COLM 2024
>
> [4] Rybkin, Oleh et al. “Value-Based Deep RL Scales Predictably.”
>
> [5] Yang, Greg. Tensor Programs V: Tuning Large Neural Networks via Zero-Shot Hyperparameter Transfer
>
> [6] McCandlish, Sam. “An Empirical Model of Large-Batch Training.”

---

### Official Review · Reviewer_qAiA · 2025-07-03

**Clarity:** 2
**Significance:** 2
**Originality:** 1
**Rating:** 4
**Confidence:** 4

**Summary:**

This paper introduces an empirical investigation of compute-optimal scaling laws for value-based deep reinforcement learning. The authors study the relationship between model size, batch size, and updates-to-data (UTD) ratio, propose empirical scaling laws (including power-law fits) for choosing these hyperparameters under compute or data budgets, and present a mechanistic explanation for “TD-overfitting” when batch size is too large for a given model size.

**Questions:**

1.	How would the proposed scaling laws change if learning rate were tuned jointly with batch size and UTD ratio? Could ignoring learning rate invalidate the power-law relationships claimed?
	2.	How robust are the observed scaling laws across different RL algorithms (e.g. SAC without BRO’s regularization, Simba, or discrete-action methods)? Is there evidence they generalize?

**Ethical Concerns:**

["NO or VERY MINOR ethics concerns only"]

**Final Justification:**

I raised my score since the authors could demonstrate the validity of their experiments including a discussion over the learning rate.

**Limitations:**

- The paper explicitly acknowledges (in Section 8) that learning rate and other hyperparameters were not included in the scaling analysis due to computational costs. This omission is non-trivial, since learning rate interacts strongly with batch size and UTD in standard practice.
- The analysis is limited to a small number of continuous control tasks with one specific algorithm variant (BRO). The empirical fits may not generalize to other domains or algorithms.
- The proposed scaling laws are based on empirical fits with relatively high error and may not provide precise prescriptions for practitioners. There is limited evidence that following these rules leads to improved final policy performance.

**Paper Formatting Concerns:**

Citation format is not coherent across the paper.

**Quality:**

2

**Strengths And Weaknesses:**

**Strenghts**

- The paper tackles an important problem in RL practice: understanding the joint scaling behavior of batch size, model size, and UTD ratio under compute constraints.
- It provides a clear empirical investigation that identifies a “TD-overfitting” phenomenon, offering intuition and diagnostics (e.g. the passive critic analysis) for why larger models tolerate larger batch sizes.
- The authors propose empirical scaling laws (power-law fits) that could be useful in guiding hyperparameter choices for practitioners under budget constraints.

**Weaknesses**

- Learning rate is completely ignored. The paper even acknowledges in its limitations (Section 8) that learning rate is an important hyperparameter not included in the scaling analysis. This is a critical omission, since when changing batch size or UTD ratio, standard practice is to adjust learning rates (e.g. via linear scaling rules or other heuristics). The claimed power-law scaling rules are therefore incomplete and of limited practical value unless learning rate effects are explicitly controlled for.
- Unclear evidence of improved performance. While the paper provides empirical fits and shows that model size and UTD ratio trade off along predicted contours, it does not clearly demonstrate that applying these scaling laws improves policy performance in practice compared to naive or baseline settings. The empirical fits are largely about validation TD-error, data efficiency metrics, and FLOPs—not about showing better returns in held-out or transfer tasks when using the proposed scaling rules.
- Primarily empirical with limited scope. Despite claiming generality, the experiments focus almost entirely on the BRO variant of SAC and use a narrow set of continuous control benchmarks (DeepMind Control Suite and HumanoidBench). There are no experiments demonstrating that the scaling rules transfer to other RL architectures (e.g. standard SAC, Simba, or discrete-action methods like PQN). This limits the generality of the proposed laws.
- Fit quality is mediocre. In multiple places the paper notes high relative error in the fits (e.g. ~50% uncertainty in batch size predictions). While they propose 2D fits to reduce grid search costs, this large error means that the scaling rules may not actually offer reliable guidance in practice.

---

> ### Author Rebuttal · Authors · 2025-07-31
>
> Thank you for the comprehensive and thoughtful response. We are glad to hear that our paper ‘provides a clear empirical investigation’ that ‘that could be useful […] for practitioners’. To address your points about improving the empirical evidence presented in the paper, we provide additional experiments to address the feedback. Due to large compute requirements and a delay in obtaining the required compute resources to rerun hyperparameter sweeps used to analyze scaling behaviors, some of our experiments are still running. We will report back and include them when they finish, within the author-reviewer discussion period.
>
> We address the remainder of the concerns below.
>
> > Learning rate is completely ignored.
>
> We have now run an additional analysis of learning rate dependencies. We find that (i) best learning rate decreases with increasing model size, correlation: -0.75, (ii) best learning rate decreases with increasing UTD, correlation: -0.46, (iii) best learning rate increases with increasing batch size, correlation: 0.42. With simple log-linear fits, we obtain a relative error of 37.5%. These results are consistent with existing literature McCandlish’18, Yang’22, Rybkin’25. We are running additional experiments to evaluate this hypothesis and will report back before the end of the discussion period.
>
> h1-crawl-v0:
> $R^2$ = 0.3417
> Relative error: 43.5702%
> $\text{lr}^* ~ 4.4827\text{e}-4 \cdot (N/2.3\text{e}6)^{-0.3112} \cdot \sigma^{-0.1273} \cdot (B/512)^{0.3709}$
>
> h1-stand-v0:
> $R^2$ = 0.5702
> Relative error: 32.2800%
> $\text{lr}^* ~ 2.4727\text{e}-4 \cdot (N/2.3\text{e}6)^{-0.2472} \cdot \sigma^{-0.2392} \cdot (B/512)^{0.2701}$
>
> Despite the high relative error, we observe that data efficiency is similar within an interval of “reasonable” learning rates, which includes our fixed value; see the “high relative error” section below. By using the optimal learning rate fit in our results, we can improve the method’s performance.  We expect this method to follow the same data efficiency law as Eq (6.1), but have better coefficients $\alpha_J, \beta_J$, because Eq (6.1) is justified with asymptotic case analysis and is consistent with results from Rybkin’25. We are currently running experiments to fit this improved data efficiency law and will report them once they finish.
>
> A complete N-dimensional analysis of learning rate was not possible in this rebuttal period due to computational requirements: 4 learning rate values x 4 batch size values x 4 model size values x 4 utd values yields 256 required GPUs per environment. However, our initial analysis provides insight into how learning rate behaves and can be used by future studies to propose more effective scaling methods.
>
> > other domains or algorithms.
>
> In addition to the two domains and 17 tasks in the original submission, we have now evaluated scaling laws on DQN in Atari. This is consistent with prior work on architectures [1,2] and RL scaling laws [3] which both showed findings generally applicable to many algorithms and environments. Interestingly, we observe that on Atari, batch size needs to be decreased with model size, which can be estimated from small-scale experiments using our workflow. We are investigating whether this is due to statistical overfitting or TD-overfitting and will report and include these results when they finish. Future work will focus on validating scaling laws on practical large-scale environments.
>
> | Model size |   $J$ = 0.16 |   $J$ = 0.27 |   $J$ = 0.38 | $J$ = 0.49   | $J$ = 0.60   |
> |-----------:|-------------:|-------------:|-------------:|:-------------|:-------------|
> |      150K  |        11880 |        19800 |        38010 | N/A          | N/A          |
> |      320K  |        10880 |        18130 |        26460 | 54130        | 82520        |
> |      790K  |        10310 |        17180 |        24050 | 40580        | 60010        |
> |     2.13M  |         8710 |        14520 |        20330 | 26800        | 36040        |
> |     6.47M  |         7860 |        13100 |        18340 | 23580        | 33270        |
>
> We ran 5 Atari-100K tasks at UTD=1, and for each $J$ estimated the data efficiency for the best batch size. Our preliminary results show that data efficiency improves with model size on average.
>
> > Unclear evidence of improved performance.
>
> In Figure 4, we show the compute-optimal settings for UTD and model size for each data budget. As we show in Appendix A.2, this scaling method is also data-optimal for each compute budget. For each compute budget $\mathcal C$, we run the configuration with UTD $\sigma^\*(\mathcal C)$, model size $N^\*(\mathcal C)$, and batch size $\tilde B(\sigma, N)$. We provide three baselines demonstrating the superior performance of compute-optimal scaling: for each compute budget, we can additionally run (i) UTD-only scaling at compute budget $\mathcal C$ for a given model size, (ii) N-only scaling at compute budget $\mathcal C$ for a given UTD, (iii) the compute-optimal UTD and model size and constant batch size (specifically, the fitted batch size from the lowest compute budget).
>
> | Aggregation | Compute optimal | UTD-only scaling | $N$-only scaling | Constant batch size |
> |-------------|-----------------|------------------|------------------|---------------------|
> | Average     | 1.00            | 1.26             | 1.11             | 1.03                |
> | Median      | 1.00            | 1.18             | 1.11             | 1.05                |
>
> In the table, we record the ratio between the data efficiency in each baseline to the compute-optimal data efficiency, aggregated over environments and compute budgets. We show that compute-optimal scaling with the fitted batch size outperforms the baselines.
>
> We further note that we expect the improvement from our approach to be more prominent as the gap between small-scale settings that we can readily experiment with and large-scale settings that we wish to find good hyperparameters for diverge from each other. For instance, tuning hyperparameters on a 1e6 parameter model and transferring to a 5e6 parameter model might be possible since the original hyperparameters are already in a good neighborhood; but transferring to a 2e9 parameter model (bare minimum for a language application) will likely work poorly unless a proper scaling law procedure is applied and hyperparameters are adjusted. As the only work establishing joint scaling laws for batch size, UTD, and compute, we believe future practitioners will rely on our results in deciding how to tune hyperparameters at large scale.
>
> > high relative error in the fits (50%)
>
> Indeed, we observe that our batch size fits have a moderate confidence interval of up to 50%. We observe this is because batch size in many cases has a wide range of values which all have near-optimal performance. Note that utilizing any of these batch values performs similarly, which means a practitioner can indeed use any value within this interval to obtain good performance and a moderate confidence interval is justified. In the table below, we group runs by UTD and model size, and bin suboptimal batch sizes. Then, we consider the data efficiency ratio between suboptimal and optimal runs, and average over UTDs and model sizes. We find that batch sizes within an interval around the best batch size $B^*$ perform reasonably, and performance degrades significantly with larger intervals. This is consistent with prior work on LLMs [4], and explains the relative error in the fit. In practice, this means that our fits, despite a large relative error, yield good results when using them to train models, which is anyways our ultimate goal.
>
> | Batch size range      | Data efficiency ratio |
> |-----------------------|-----------------------|
> | [1/16 B*, 1/8 B*]     | 1.52                  |
> | [1/8 B*, 1/4 B*]      | 1.38                  |
> | [1/4 B*, 1/2 B*]      | 1.26                  |
> | [1/2 B*, 2/3 B*]      | 1.22                  |
> | [1.5B*, 2B*]          | 1.16                  |
> | [2B*, 4B*]            | 1.18                  |
> | [4B*, 8B*]            | 1.19                  |
> | [8B*, 16B*]           | 1.30                  |
>
> Similarly, we observe that there is a range of reasonable learning rates:
>
> | Learning rate range   | Data efficiency ratio |
> |-----------------------|-----------------------|
> | [1/4 lr*, 1/2 lr*]    | 1.39                  |
> | [1/2 lr*, 2/3 lr*]    | 1.35                  |
> | [2/3 lr*, lr*]        | 1.04                  |
> | [lr*, 1.5 lr*]        | 1.03                  |
> | [1.5 lr*, 2 lr*]      | 1.18                  |
> | [2 lr*, 4 lr*]        | 1.27                  |
>
> In practice, we ran our main experiments at a constant learning rate of 3e-4. Running an additional sweep over learning rates of [1e-4, 2e-4, 6e-4] showed that only 2e-4 had superior performance in some cases. Since we are within the [lr*, 1.5 lr*] interval, the overall effect on performance is minimal.
>
> We believe the submission is now much stronger after addressing your feedback. Please let us know if there is anything else in the submission that you think we should address for an improved score.
>
> [1] Nauman, Michal, et al "Bigger, Regularized, Optimistic." NeurIPS 2024
>
> [2] Lee, Hojoon, et al. "Simba: Simplicity bias for scaling up parameters in deep reinforcement learning." ICLR 2025
>
> [3] Rybkin, Oleh et al. “Value-Based Deep RL Scales Predictably.”
>
> [4] Grattafiori, Aaron et al. “The Llama 3 Herd of Models.”
>
> [5] McCandlish, Sam. “An Empirical Model of Large-Batch Training.”
>
> [6] Yang, Greg. Tensor Programs V: Tuning Large Neural Networks via Zero-Shot Hyperparameter Transfer

---

> > ### Author Response · Authors · 2025-08-08
> >
> > Thank you for review and feedback once again! We have now completed some of the additional runs we promised in the rebuttal and would be grateful if you are able to take a look at the rebuttal and the additional responses below to let us know if your concerns are addressed.
> >
> > **Please let us know if you find your concerns addressed, and if so, we would be grateful if you are willing to adjust your score.**
> >
> > > Learning rate
> >
> > In the original rebuttal response, we reported a learning rate fit within a grid search setting, to obtain a rough estimate for the best learning rate in terms of UTD, model size, and batch size. We also identified that, within the grid search regime, the data efficiency is not very sensitive to changes in the learning rate.
> >
> > We have now additionally run experiments verifying this hypothesis about sensitivity to the learning rate in the compute-optimal setting, where we have larger model sizes and UTDs. For each task and compute-optimal setting $(N, \sigma, B)$, we ran a sweep of learning rates over (1e-4, 2e-4, 3e-4, 4e-4, 5e-4; default 3e-4). Following the same bootstrapping procedure as for the batch size, we can estimate, for each setting, the best learning rate. Then, we can estimate the corresponding data efficiency by rounding to the nearest searched learning rate.
> >
> > | Environment    | Data efficiency ratio |
> > |:---------------|----------------------:|
> > | h1-crawl-v0    | 1.0118                |
> > | h1-pole-v0     | 1.0000                |
> > | h1-stand-v0    | 1.0000                |
> > | humanoid-stand | 0.9504                |
> >
> > Taking the mean over all settings, the compute-optimal data efficiency is improved by merely 1.2% by using these optimal fit learning rates. This is very small – meaning that the best learning rates do not meaningfully improve performance, and a reasonable learning rate is sufficient for the compute-optimal setting. (Note here that compute-optimal settings can include higher model sizes and UTDs, i.e. up to width 3024 and up to UTD 18, depending on the task.) Of course, we should be careful in not over generalizing this observation to every setting, but it confirms that utilizing learning rate does not change the functional form of our data efficiency and optimal compute fits. That said, we think discussing the learning rate fits makes our paper more complete and we will add it to the final version of the paper.
> >
> > > Atari 100k
> >
> > We are still running some of the Atari100k results. While we have completed runs for 5 seeds on 10 environments (Alien, Amidar, Assault, Asterix, Breakout, Freeway, Krull, MsPacman, Pong, Qbert, Seaquest), we are still observing a substantially large confidence interval on most of our aggregate runs. This is consistent with RLiable (Agarwal et al. 2022), which argues that confidence intervals and seed variance can be extremely large on this benchmark. In order to conclude anything significant and concrete about this task, we will need a way to run more seeds and more environments, and do not want to prematurely come to a conclusion. We are continuing to run these experiments and will add them to the final version of the paper.

---

### Official Review · Reviewer_9ANn · 2025-07-04

**Clarity:** 3
**Significance:** 3
**Originality:** 3
**Rating:** 4
**Confidence:** 4

**Summary:**

The paper investigates scaling laws in deep RL, focusing on the batch size and model capacity (layer widths). It is found that increasing the batch size only improves performance when model capacity is sufficently large but not when model capacity is small. A phenomenon called "TD overfitting" is posited, where using small networks produces larger TD errors on validation data when batch size is increased. The trend is reversed with larger netweorks.  Experiments are run with the BRO algorithm and continuous control benchmark tasks.

**Questions:**

Main questions:
- The part starting at line 155 "A mental model for TD-overfitting": While the explanation makes intuitive sense, there are no experiments to support this specific mechanism. The idea seems to be related to having less interference between updates when the model is larger, it would be nice to investigate this hypothesis directly. For example, meausuring the amount of change in the value function prediction at validation states given small and large models.
Also, the findings in [1] could be relevant here, where they find that using TD updates seems to lead to more decoupled representations.

- It would seem like the target network update rate (for soft updates) or update frequency (for hard updates) would be an important factor affecting TD overfitting. In particular, it controls how fast the targets change which may induce more nonstationarity in the learning process. In line 180, nonstationarity is mentioned as a key part of the mechanism explaining TD overfitting. Have you explored this direction? Some investigations would be a great addition. This could be done with a single UTD ratio or other simplifying assumptions, given the computational costs.

- Do you expect similar findings to hold for different learning algorithms such as Q-learning or PPO? The "critic" is used in a different way in those methods comapared to SAC.  (I do not expect additional experiments for this, just your opinion).

Clarification questions:
These impacted my understanding of the paper so answer to these would help properly evaluate it.
- Equation 5.1: The notation is not clearly explained. What does $\alpha$ and $\beta$ refer to and how are the subsc

- Fig. 3 shows curves fit to only 4 data points each. If there are 2 free variables in the equation, it's difficult to validate that the fitted curve is of the correct form. Many equations could fit reasonably well. It is mentioned in the text that a range of batch sizes work well for some of these settings. In that case, perhaps a heatmap/scatterplot similar to Fig. 4 with batch size vs. UTD on the axes and colors for the performance value could work better. We would be able to see which regions are good. Stars could denote the points with optimal performance for each UTD or model size.

- In Fig. 4: How are the contours generated? Clearly, a continuum of parameters are not tested so there must be some method to interpolate and draw the curves.

- Is the actor network also scaled up with the model size or is it only the critic? It seems like TD overfitting would be an issue mainly relevant to the critic.



[1] "Interference and Generalization in Temporal Difference Learning" Bengio et al.

**Ethical Concerns:**

["NO or VERY MINOR ethics concerns only"]

**Limitations:**

These are discussed.

**Quality:**

3

**Strengths And Weaknesses:**

Generally, scaling deep RL agents is still challenging and improving our understanding is an interesting direction. As such, the paper tackles a topic releavnt to the community.

The presentation of the paper is nice and generally clear. They are some issues with unexplained notation and some parts to be clarified (as outlined in the Questions section).

The experimental findings about how model capacity can be an important factor to unlock batch size scaling is intriguing and fits well with results from prior works showing that scaling the critic is helpful. It's interesting to see another way that TD learning may lead to different learning dynamics than supervised learning, similar to previous work [1,2].
I find this part to be the most interesting contribution of the paper and I would preferred a bit more investigation into the TD overfitting phenomena (see Questions).

I am leaning towards acceptance currently and would be happy to raise my score given proper responses to the questions.

[1] "Learning Dynamics and Generalization in Reinforcement Learning" Lyle et al.
[2] "Interference and Generalization in Temporal Difference Learning" Bengio et al.

---

> ### Author Rebuttal · Authors · 2025-07-31
>
> Thank you for the thorough and engaging feedback, and for a positive assessment of our work. We believe that the alternate perspectives and references that you discuss are quite relevant, and we will discuss them in the updated version of the paper.  We discuss your concerns below and we have also performed additional experiments to address some of the concerns.
>
> >  interference between updates
>
> Thanks for pointing out this paper, we will cite it and discuss it in the updated version. Briefly, Bengio’20 discusses a phenomenon where TD learning leads to low interference due to a lack of temporal coherence in the value network. This is consistent with our analysis: a lack of temporal coherence suggests a poor target distribution, which in turn yields overfitting. A more detailed study of this phenomenon is warranted. However, a complicating factor is that measuring value changes can be challenging in continuous environments, and absolute value changes might not correspond to meaningful updates in the parameter space. We will perform proof-of-concept experiments on this and include them in a later version.
>
> > target network update rate
>
> Intuitively, target update rate should be important for the TD-overfitting phenomenon. In our preliminary experiments, we found that when holding other hyperparameters constant, TD-overfitting increases when setting the target update rate too high. That said, we saw that the effect of the target network update rate was overshadowed by setting a different learning rate and running training at a different UTD value & batch size. In other words, our initial experiments indicated that instead of changing the target network update rate we could find an alternate configuration of other hyperparameters that gave the same performance, and the target update rate was less necessary to account for.
>
> Given a limited amount of computational resources, we had to make a conscious decision early on in the project regarding which hyperparameters to focus on and given the above preliminary findings, we decided to skip the target update rate. A full study of such effects is an exciting direction of future work, and we will rerun some of the preliminary experiments into the appendix of the final version of the paper. We do agree that there might be some nuanced effects of the target network update rate on TD-overfitting, which would be good to study in future.
>
> > Do you expect similar findings to hold for different learning algorithms
>
> We ran 5 Atari-100K tasks with DQN at UTD=1, and for each $J$ estimated the data efficiency for the best batch size. Our preliminary results show that data efficiency improves with model size on average.
>
> | Model size |   $J$ = 0.16 |   $J$ = 0.27 |   $J$ = 0.38 | $J$ = 0.49   | $J$ = 0.60   |
> |-----------:|-------------:|-------------:|-------------:|:-------------|:-------------|
> |      150K  |        11880 |        19800 |        38010 | N/A          | N/A          |
> |      320K  |        10880 |        18130 |        26460 | 54130        | 82520        |
> |      790K  |        10310 |        17180 |        24050 | 40580        | 60010        |
> |     2.13M  |         8710 |        14520 |        20330 | 26800        | 36040        |
> |     6.47M  |         7860 |        13100 |        18340 | 23580        | 33270        |
>
>
> We believe Atari is more susceptible to overfitting due to the deterministic nature of the environments. However, in general, since DQN also relies on TD-learning, we expect similar trends to apply. In contrast, PPO, when run with standard hyperparameter settings, relies on policy gradients more than TD-learning. Therefore, we expect that PPO scaling laws will be more similar to supervised learning scaling laws and simpler. Studying whether PPO is also affected by the TD-overfitting phenomenon we describe is an interesting direction of future work, particularly for applications using policy gradient methods, such as LLMs.
>
> > The notation is not clearly explained
>
> We use $\alpha, \beta$ for the power coefficients and $a, b$ for multiplicative coefficients. The subscripts specify the equation that the terms appear in, e.g. $a_B$ in $B(., .)$, $a_J$ in $D_J(. ,.)$, $a_C$ in $C^*(.)$, etc.
>
> > Fig. 3 shows curves fit to only 4 data points each
>
> We will provide heatmaps in the appendix. Note that the fit in Eq (5.1) has 4 variables, and is fit to 16 points. All 16 points are shown in Fig 3. We have considered different functional forms and selected this one as it had the best empirical performance and intuitive justification as discussed on l223. Specifically, with large $\sigma$, it will dominate the denominator, requiring a small batch size, consistent with Rybkin’25. With small $\sigma$, the model size term will dominate, consistent with Kaplan’20.
>
> > In Fig. 4: How are the contours generated?
>
> We plot predictions from the fit (6.1). Because the fit is two-dimensional (it depends on $\sigma$ and N), we are able to produce a continuum of predictions, which is necessary for proper extrapolation.
>
> > Is the actor network also scaled up
>
> In line with prior work [1], we found that scaling critic-only leads to best performance.
>
> Please let us know if there is anything else in the submission that you think we should improve or clarify for an improved score.
>
> [1] Nauman, Michal, et al "Bigger, Regularized, Optimistic." NeurIPS 2024
>
> [2] Rybkin, Oleh et al. “Value-Based Deep RL Scales Predictably.”
>
> [3] Kaplan, Jared, et al. “Scaling Laws for Neural Language Models”. arXiv 2020

---

> > ### Comment · Reviewer_9ANn · 2025-08-06
> >
> > Thank you for the clarifications, they are satisfactory.
> > After having read the rebuttal and other reviews, there are still some clear limitations to the work e.g. using BROnet only, having to omit certain variables such as target net frequency or n-step TD from the analysis and relatively large errors on some scaling law fits. I think some investigation of the TD and interference interactions would also be a great addition.
> > Overall, I will maintain my current score since I believe there are still useful contributions.

---

### Note · Authors · 2025-08-13

Dear AC and Reviewers,

Thank you for your feedback and discussions. Three of the reviewers are positive about the paper. We believe the paper improved substantially through the discussion period, since we could add several experiments that show interesting conclusions and support the claims in the paper. We briefly summarize these findings below:

* Our compute-optimal tradeoff between UTD and model size outperforms UTD-only scaling, model-size-only scaling, and constant batch size scaling.
* Our batch size fit enables significant data efficiency improvements in the grid search regime. Within a “reasonable” range, data efficiency is not very sensitive to the batch size, enabling us to use several values of batch size with minimal degradation in performance. This appears as a large relative error in our batch size fit as we clarified to reviewers, but these errors are benign.
* In the paper (Figure 3), we saw that the best batch size can vary by more than an order of magnitude even when we only vary the model size at a constant UTD. A constant batch size would fall outside of this “reasonable” range and significantly worsen data efficiency scaling.

We found that varying other hyperparameters did not strongly affect data efficiency but could be an exciting direction for future work.

* In the grid search regime of (UTD, model size), within a “reasonable” range, data efficiency is not very sensitive to learning rate. The best learning rate is never more than 50% away from our “default”, and is always within the reasonable range.
* In the compute-optimal regime including higher compute budgets, we additionally show that the best learning rate outperforms our “default” learning rate by only 1.2% on average.
* We find that $\tau$, the target network update rate, can vary by over an order of magnitude with little effect on data efficiency.

We will also add a discussion of several other aspects that we could not study (primarily due to computational limitations) in this work. We believe our work should serve as a stepping stone towards fully resolving questions around the science of scaling for value-based deep RL, which is becoming critical to study as RL continues to be used at scale in several downstream fields (e.g, LLMs, VLMs, etc).

We hope that the clarifications and the discussion will lead the reviewers and AC to conclude that the paper is valuable to be accepted. We also look forward to future progress in the direction of scaling up value-based RL.

---

### Decision · Program_Chairs · 2025-09-17

**Decision:**

Accept (poster)

**Comment:**

a) Scientific Claims and Findings: The paper investigates compute-optimal scaling for value-based deep RL. They charactrize TD-overfitting as small models perform poorly with large batch sizes while larger models benefit from them. The authors develop empirical scaling laws for optimally allocating compute between model size and updates-to-data ratio.

b) Strengths: The work addresses a critical scaling problem with novel theoretical insights that reconcile conflicting batch size practices across RL domains. The TD-overfitting hypothesis provides compelling mechanistic explanations supported by comprehensive empirical investigations and delivers practical scaling guidance.

c) Weaknesses and Missing Elements: Primary limitations include narrow focus on BRO architecture raising generalizability concerns, initial omission of learning rate analysis, and high relative errors in scaling law fits. The experimental scope was limited to continuous control with insufficient validation on discrete action spaces.

d) Decision Rationale: The AC recommends the paper for acceptance due to its novel TD-overfitting contribution and practical scaling guidance for an increasingly important problem. The authors demonstrated extensive additional experiments that substantially strengthened the paper.

e) Rebuttal Discussion Summary: Reviewers raised concerns about learning rate omission, mechanistic understanding, and architectural limitations. Authors addressed these through comprehensive learning rate analysis, temporal coherence experiments, and additional DQN results, leading three reviewers to maintain or raise scores.